# Transsaccadic working memory in healthy ageing and neurodegenerative disease

**Sijia Zhao[1]\*[†], Thomas Parr[2†], Rob Udale[1], Verena Klar[1], Gabriel Davis Jones[2,3], Anna Scholcz[1,2], Sofia Toniolo[2,4], Sanjay G Manohar[1,2,4], Masud Husain[1,2,4]**

[1]Department of Experimental Psychology, University of Oxford, Oxford, United Kingdom; [2]Nuffield Department of Clinical Neurosciences, University of Oxford, Oxford, United Kingdom; [3]Nuffield Department of Women's and Reproductive Health, University of Oxford, Oxford, United Kingdom; [4]Cognitive Disorders Clinic, John Radcliffe Hospital, University of Oxford, Oxford, United Kingdom

## eLife Assessment

This study makes an **important** contribution by revealing how saccades selectively disrupt spatial working memory while sparing other object features, and by demonstrating how this mechanism is altered in aging and neurodegeneration. The findings are supported by **convincing** evidence derived from well-controlled eye-tracking experiments and systematic generative model comparisons. Together, the work provides a computationally grounded framework that is of importance for understanding trans-saccadic memory and its clinical relevance.

**\*For correspondence:**
sijia.zhao@psy.ox.ac.uk

[†]These authors contributed equally to this work

**Competing interest:** The authors declare that no competing interests exist.

**Abstract** The brain continuously integrates rapidly changing visual input across eye movements to maintain stable perception, yet the precise mechanisms underpinning dynamic working memory and how these break down in brain diseases remain unclear. We developed a novel eye-tracking paradigm and computational models to investigate how spatial and colour information are updated across saccades in the human brain. Our findings reveal that saccades selectively impair spatial but not colour memory. Computational modelling identified that spatial representations are maintained in a dual eye-centred frame of reference which is actively updated by a noisy memory of saccades but is vulnerable to interference. Using this model, we found that specific mechanistic failures in initial encoding and memory decay, rather than the saccadic updating process itself, account for spatial working memory deficits in Alzheimer's and Parkinson's disease. These results provide a mechanistic understanding of how dynamic spatial memory operates in health and its disruption in neurodegenerative disorders.

## Introduction

Natural vision is profoundly dynamic, relying upon sequences of saccades. Everyday tasks, such as making a cup of tea (*Land and Hayhoe, 2001*; *Crawford et al., 2004*), arranging the layout of objects (*Draschkow et al., 2021*; *Ballard et al., 1995*), or copying a simple line drawing (*Cohen, 2005*; *McManus et al., 2010*; *Perdreau and Cavanagh, 2015*), require us to precisely track spatial locations—which appear at different retinotopic locations with each fixation—across saccades (*Grzeczkowski et al., 2023*). This requires a dynamic updating process, termed 'remapping', that involves updating our internal representation of object locations to maintain a stable world-centred frame of reference (*Duhamel et al., 1992*; *Colby and Goldberg, 1999*; *Golomb and Mazer, 2021*; *Melcher and Colby, 2008*). One key hypothesis is that we use working memory across visual fixations to update perception dynamically (*Bays and Husain, 2007*; *Morris and Krekelberg, 2019*; *Harrison et al.,*

*2024*). However, despite the apparent stability of our visual world, it has proven difficult to separate the distinct contributions of retinotopic memory from transsaccadic memory. To precisely characterise the interplay of factors contributing to dynamic spatial working memory—particularly amidst the complexities of multiple fixations, as well as the potential for memory decay and interference—and to move beyond purely descriptive observations, fundamental questions regarding the underlying mechanisms and sources of potential errors need to be addressed.

Central to this enquiry is the nature of the coordinate system used for the brain's internal spatial representation. Does the brain maintain a single, world-centred (allocentric) map, or does it rely on a dynamic, eye-centred (retinotopic) representation (*Golomb and Mazer, 2021*; *Bays and Husain, 2007*; *Irwin, 1992*; *Golomb, 2019*)? In the latter system, retinotopic memory preserves spatial information within a fixation, whereas transsaccadic memory describes the active process of updating these representations across eye movements to achieve spatiotopic stability—the perception of a stable world despite eye movements (*Golomb and Mazer, 2021*; *Golomb, 2019*; *Irwin, 1991*; *Cavanagh et al., 2010*). If spatial stability is indeed reconstructed through such remapping, the mechanism remains unresolved: do we retain memories of absolute fixation locations, or do we reconstruct these positions from noisy memories of the intervening saccade vectors? We can test these hypotheses by analysing when and where memory errors occur. Assuming that memory precision declines over time (*Schneegans and Bays, 2018*), the resulting error distributions should reveal the specific variables that are represented and updated across each saccade.

The clinical relevance of these spatial mechanisms is underscored by significant disruptions to visuospatial processing and constructional apraxia—a deficit in copying and drawing figures—observed in neurodegenerative conditions such as Alzheimer's disease (AD) and Parkinson's disease (PD) (*Cormack*

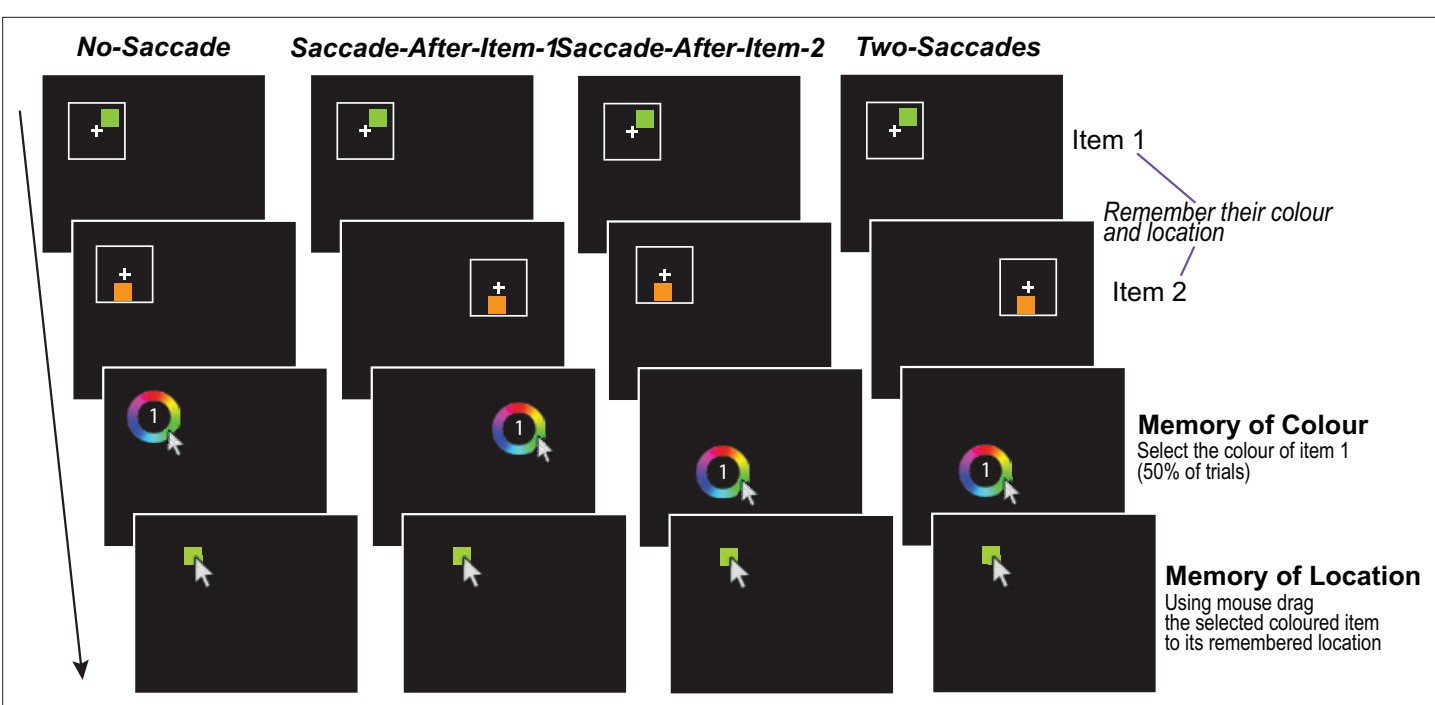

**Figure 1.** Location and colour updating across saccades (LOCUS) task. Participants were asked to fixate on a white cross wherever it appeared. They had to remember the colour and location of a sequence of two briefly presented coloured squares (Items 1 and 2), each appearing within a white square frame. They then fixated a colour wheel wherever it appeared on the screen, which served as the target for the second instructed saccade (i.e. a movement from the second fixation cross to the colour wheel location). This cued recall of a specific square (Item 1 or Item 2 labelled within the colour wheel). Participants selected the remembered colour on the colour wheel, which led to a square of that colour appearing on the screen. They then dragged this square to its remembered location on the screen. Saccadic demands were manipulated by varying the locations of the second frame and the colour wheel, resulting in four conditions in their reliance on retinotopic vs. transsaccadic memory: (1) No-Saccade condition providing a baseline measure of within-fixation precision as no eye movements were required. (2) Saccade After Item 1; (3) Saccade After Item 2; (4) Saccades after both items (Two Saccades condition). In all conditions requiring eye movements, saccade vectors were constrained to a minimum amplitude of 8.5° (degrees of visual angle). While the No-Saccade condition isolates retinotopic working memory, conditions (2) to (4) collectively quantify the impact of varying saccadic demands and timings on the maintenance of spatial information, thereby assessing the efficacy of the transsaccadic updating process.

*et al., 2004*; *Trojano and Gainotti, 2016*). This raises a crucial question: Do clinical impairments in complex visuomotor tasks stem from specific failures in transsaccadic remapping? If so, the computational parameters that define normal spatial updating should also provide a mechanistic account of these clinical deficits, differentiating them from general age-related decline.

To address these questions and move beyond purely descriptive observations towards mechanistic models, we developed a novel eye-tracking paradigm—the location and colour updating across saccades (LOCUS) paradigm (*Figure 1*). This paradigm allows us to directly compare retinotopic and transsaccadic working memory for both spatial location and colour. Combined with computational modelling, our approach allowed us to precisely characterise the interplay of factors contributing to dynamic spatial working memory, isolate the impact of saccades, and test specific, competing hypotheses regarding transsaccadic memory. Including young and older healthy adults, as well as patients with PD and AD, our findings reveal that saccades selectively disrupt spatial but not colour memory.

Across all healthy and patient groups, the winning computational model indicated that spatial representations are maintained in a dual (retinotopic) reference frame, updated based on a noisy memory of the saccade vector, and susceptible to interference from other items in memory. Importantly, we identified specific model parameters—related to angular encoding, memory decay, and interference—that differentiated the patient groups from healthy controls, providing a mechanistic account of their visuospatial deficits. Finally, by linking these mechanistic parameters to a standard clinical measure of constructional ability (the Rey-Osterrieth complex figure [ROCF] task), we demonstrate that transsaccadic updating represents a core computational phenotype underpinning real-world visuospatial construction in both health and neurodegeneration.

## Results
### Saccades disrupt spatial memory but spare colour
In the LOCUS paradigm (*Figure 1*), participants memorised the colour and location of two sequentially presented squares. Subsequently, they were cued to recall the colour and location of one of these squares, i.e., either the first or the second. Colour recall involved clicking on a colour wheel to indicate the remembered colour using the mouse. Location recall required dragging a colour-matched square to the remembered position on the screen.

By manipulating the spatial location of the squares and also that of the colour wheel, we controlled saccadic demands on every trial, creating four conditions with equal probability: no-saccade, saccade after Item 1, saccade after Item 2 (to the colour wheel), or a saccade after both items. The no-saccade trials provided a measure of retinotopic working memory, while the two-saccade trials assessed transsaccadic working memory.

To mitigate potential confounds, we monitored eye position throughout the experiment. Eye-tracking analysis confirmed high compliance in healthy adults, who followed instructions on the vast majority of trials (younger adults: 97.2 ± 5.2%; older adults: 91.3 ± 20.4%). The mean difference between these groups was negligible, representing just 1.25 trials per condition, and was not statistically significant ($t(80) = 0.16$, p=1.000; see more in Materials and methods—*Eye-tracking data analysis*). Non-compliant trials were excluded from all further analyses.

This design allowed us to isolate and quantify the unique impact of saccades on spatial memory, enabling us to test competing hypotheses regarding spatial representation. If spatial memory were solely underpinned by an allocentric mechanism, precision should remain comparable across all conditions as the representation would be world-centred and unaffected by eye movements. Thus, performance in the no-saccade condition should be comparable to the two-saccade condition. Conversely, if spatial memory relies on a retinotopic representation requiring active updating across eye movements, the two-saccade condition was anticipated to be the most challenging due to cumulative decay in the memory traces used for stimulus reconstruction after each saccade (*Golomb and Kanwisher, 2012*). Critically, we hypothesised that this saccade cost would be specific to the spatial domain; while location requires active remapping via noisy oculomotor signals, non-spatial features like colour are not inherently tied to coordinate transformations and should therefore remain stable (see more in Discussion below).

Meanwhile, the no-saccade condition was expected to yield the most accurate localisation, relying solely on retinotopic information (retinotopic working memory). These predictions were confirmed

in young healthy adults ($N$=21, mean age = 24.1 years, ranged between 19 and 34; *Figure 2A and B*). A repeated-measures ANOVA revealed a significant main effect of saccades on location memory ($F$(3,20)=51.52, p<0.001, partial $\eta^2$=0.72), with Bayesian analysis providing decisive evidence for the inclusion of the saccade factor ($BF_{incl}$ = 3.52 × $10^{13}$, p(incl|data)=1.00). In contrast, colour memory remained remarkably stable across all saccade conditions ($F$(3,20)=0.57, p=0.64, partial $\eta^2$=0.03). This null effect was supported by Bayesian analysis, which provided moderate evidence in favour of the null hypothesis ($BF_{01}$=8.46, p(excl|data)=0.89), indicating that the data were more than eight times more likely under the null model than a model including saccade-related impairment.

This 'saccade cost'—the loss of memory precision following an eye movement—indicates that spatial representations require active updating across saccades rather than being maintained in a static, world-centred reference frame.

Critically, our comparison between spatial and colour memory does not rely on the absolute magnitude of errors, which are measured in different units (degrees of visual angle vs. radians). Instead, we assessed the relative impact of the same saccadic demand on each feature within the same trial. While location recall showed a robust saccade cost, colour recall remained statistically unchanged. To ensure this null effect was not due to a lack of measurement sensitivity, we examined the recency effect; recall performance for the second item was predicted to be better than for the first stimulus in each condition (*Broadbent and Broadbent, 1981*; *Murdock, 1962*). As expected, colour memory for Item 2 was significantly more accurate than for Item 1 ($F$(1,20) = 6.52, p=0.02, partial $\eta^2$=0.25), demonstrating that the task was sufficiently sensitive to detect standard working memory fluctuations despite the absence of a saccade-induced deficit.

Regarding the effect of item order on location error, saccades affected both items similarly, with no interaction between saccades and item ($F$(2.6,52.7)=0.64, p=0.57, partial $\eta^2$=0.03). However, as expected, spatial memory for Item 1 was significantly less accurate than for Item 2 ($F$(1,20) = 12.5, p=0.002, partial $\eta^2$=0.38). Crucially, even a single saccade after Item 2 significantly reduced location memory accuracy compared to the no-saccade condition ($t$=−7.64, p(bonf)<0.001).

Moreover, larger saccades were associated with increased spatial memory disruption (*Figure 2— figure supplement 1*). A multiple linear regression predicting location error showed that only the vertical distance between stimulus frames significantly predicted error ($t$(41) = 5.73, p<0.0001), with marginal significance for horizontal saccade direction ($t$(41) = −1.94, p=0.059). This suggests that greater displacement, particularly vertical, exacerbates spatial memory impairment, irrespective of saccade direction. However, there was no evidence that crossing saccades (e.g. rightward then leftward) or specific horizontal/vertical saccade directions differentially affected spatial memory precision, as revealed by repeated-measures ANOVAs ($F$(1.3,49.8)=0.04, p=0.89, partial $\eta^2$=0.001 for horizontal crossing; $F$(1.7,62.8)=0.29, p=0.71, partial $\eta^2$=0.008 for vertical crossing).

Further analysis examined whether individual differences in baseline memory precision (no-saccade condition) predicted resilience to saccadic disruption. Crucially, individual saccade costs (defined as the precision loss relative to baseline) did not correlate with baseline precision (rho = 0.20, p=0.20). This suggests that the noise introduced by transsaccadic remapping acts as an independent, additive source of variance that is not modulated by an individual's underlying memory capacity. These findings imply a functional dissociation between the mechanisms responsible for maintaining a representation and those involved in its coordinate transformation.

## Saccadic interference with spatial memory is age-independent

To investigate the impact of ageing, we replicated the experiment in a group of healthy adults above 60 (Elderly Healthy Adults; $N$=21; mean age = 72.4 years, ranged between 60 and 80). A three-way ANOVA (4 saccade conditions × 2 items × 2 age groups) revealed a main effect of age, with older participants demonstrating lower overall accuracy compared to their younger counterparts (*Figure 2A*, F(1.48,20)=8.38, p=0.003, partial $\eta^2$=0.30, Greenhouse-Geisser sphericity correction applied), with Bayesian analysis providing decisive evidence for the inclusion of the saccade factor ($BF_{incl}$ = 251.4, p(incl|data)=1.00). This age-related decline was evident in both transsaccadic working memory (Two-Saccades Location Error, rho = 0.57, p<0.001, Fisher's z=0.64; partial rho = 0.55, p<0.001, z=0.62 after controlling gender and ACE total score) and retinotopic working memory (No-Saccade Location Error, rho = 0.70, p<0.001, Fisher's z=0.86; partial rho = 0.69, p<0.001, z=0.85).

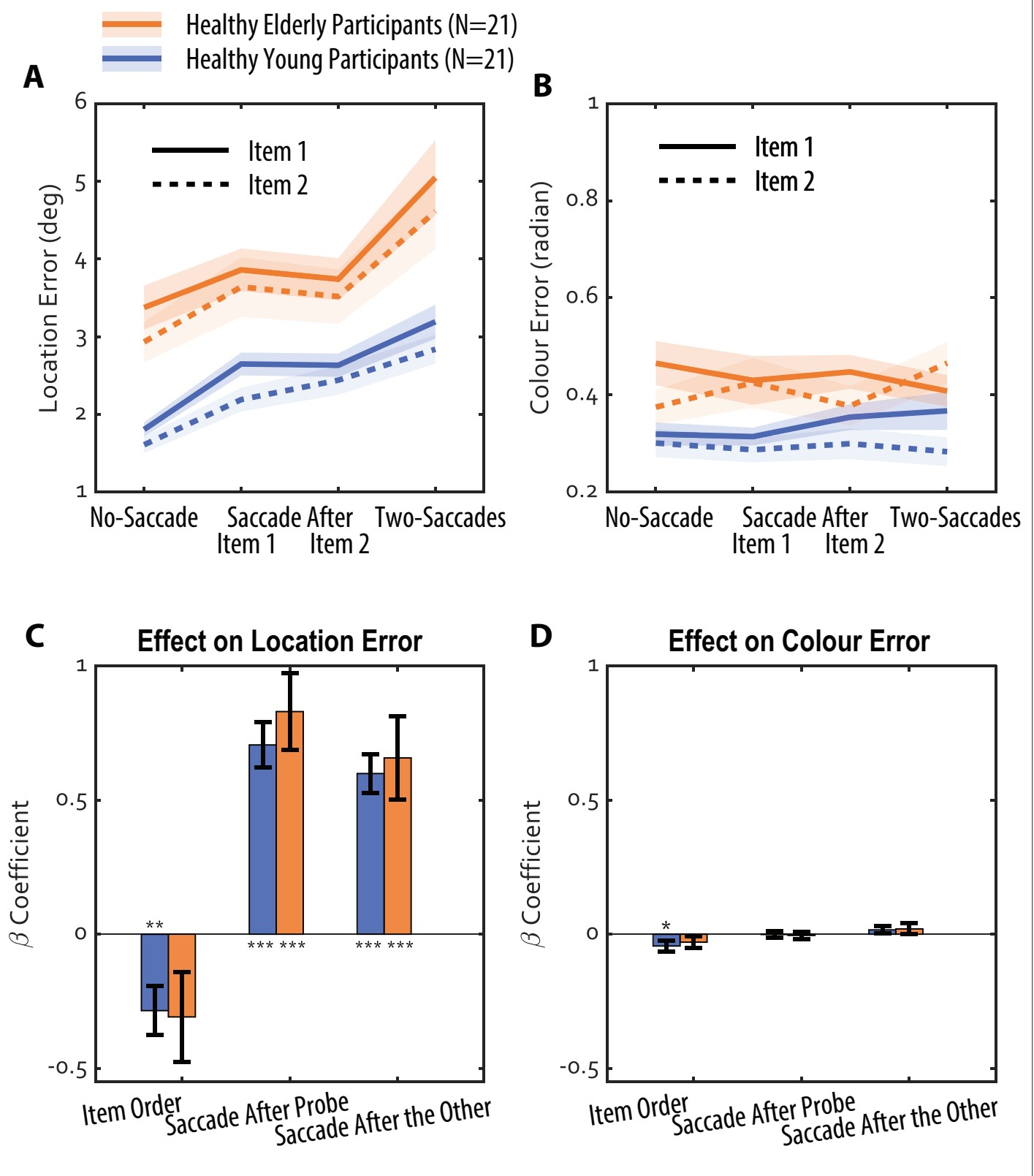

**Figure 2.** Saccades lead to increased recall error for location but not colour. (**A**) Mean location error for each saccade condition, and for each probed item (Item 1: solid lines; Item 2: dashed lines). Data are shown for young (*N*=21) and elderly (*N*=21) healthy participants. (**B**) Colour error for the same conditions. Shaded error represents ±1 standard error from mean. (**C**) Beta coefficients from a multiple linear regression model predicting location error, illustrating the effect of item order (1 or 2), saccade after the probed item, and saccade after the other item, for young (blue) and elderly (orange)

*Figure 2 continued on next page*

*Figure 2 continued*

participants. Error bars represent ± 1 standard errors. (**D**) The same model applied to colour errors, revealing a recency effect but no impact of saccade demands. ** means significantly different from zero p<0.01, *** means significantly different from zero p<0.001.

The online version of this article includes the following figure supplement(s) for figure 2:

**Figure supplement 1.** Effect of saccade direction and distance on transsaccadic memory.

---

In contrast, colour memory remained unaffected by saccade demands ($F(3,20)=0.57$, p=0.65, partial $\eta^2=0.03$), again supported by the Bayesian analysis: $BF_{01}=8.68$, p(excl|data)=0.90.

These findings suggest that while both retinotopic and transsaccadic working memory decline with age, the relative impact of saccades on spatial memory remains stable across age groups. This was also supported by the lack of significant interaction between age group, saccade condition, and item type (all p>0.23), suggesting that the detrimental effect of saccades on spatial memory is constant across both young and older adults. This observation was further supported by post hoc tests, which confirmed the *same* pattern of location errors across saccade conditions in both groups: the no-saccade condition exhibited the least location error ($t(20) \geq 5.94$, p(bonf)<0.001), followed by the one-saccade conditions (no difference between them: $t(20) = 0.02$, p(bonf)=1.00, Cohen's $d=0.002$), and with the highest error on the two-saccade condition ($t(20) \geq 3.53$, p(bonf)$\leq$0.006).

A multiple linear regression model, incorporating data from both age groups, further corroborated these findings. Both earlier item presentation and the presence of saccades after either item predicted increased location error (***Figure 2C***) but not colour error (***Figure 2D***). There was no difference in the beta coefficients between the two age groups.

In summary, while healthy ageing was associated with a general decline in spatial memory accuracy, the specific disruptive effect of saccades on spatial—but not colour—memory remained constant across age groups. Thus, the mechanisms underlying transsaccadic remapping and its impact on spatial working memory are generally preserved in healthy ageing, despite an overall reduction in memory performance.

## Computational model comparison reveals the mechanisms of transsaccadic memory

To dissect the specific cognitive processes underlying the observed saccade cost on spatial memory, we turned to computational modelling with a suite of seven computational models designed to test specific, competing hypotheses about how the brain represents and updates visual information across saccades (***Table 1***). The models are organised hierarchically, starting from a simple, single-representation model and progressively incorporating more complex mechanisms. This allows us to systematically identify which mechanisms are necessary to account for the observed behaviour. The model, data, and analysis scripts are all available in the open dataset: https://osf.io/95ecp/.

Our modelling framework addresses three core conceptual distinctions:

1. **Reference frame**. Whether spatial memory is maintained in a single, world-centred (allocentric) map (***Figure 3A***), or relies on a dual, fixation-centred (retinotopic) reference frame (***Figure 3B***).
2. **Distractor interference**. Whether memory is susceptible to interference from other items held in memory (***Figure 3C***).

**Table 1.** Hypotheses expressed as seven competing models.

| Index | Model name | Components |
|---|---|---|
| 1 | Allocentric | Allocentric Encoding Error+Allocentric Decay |
| 2 | Dual | Radial Encoding Error+Radial Decay+Angular Encoding Error+Angular Decay |
| 3 | Dual (Fixation) | Dual+Fixation Decay |
| 4 | Dual (Saccade) | Dual+Saccade Encoding Error+Saccade Decay |
| 5 | Dual+Interference | Dual+Interference |
| 6 | Dual (Fixation)+Interference | Dual+Fixation Decay+Interference |
| 7 | Dual (Saccade)+Interference | Dual+Saccade Encoding Error+Saccade Decay+Interference |

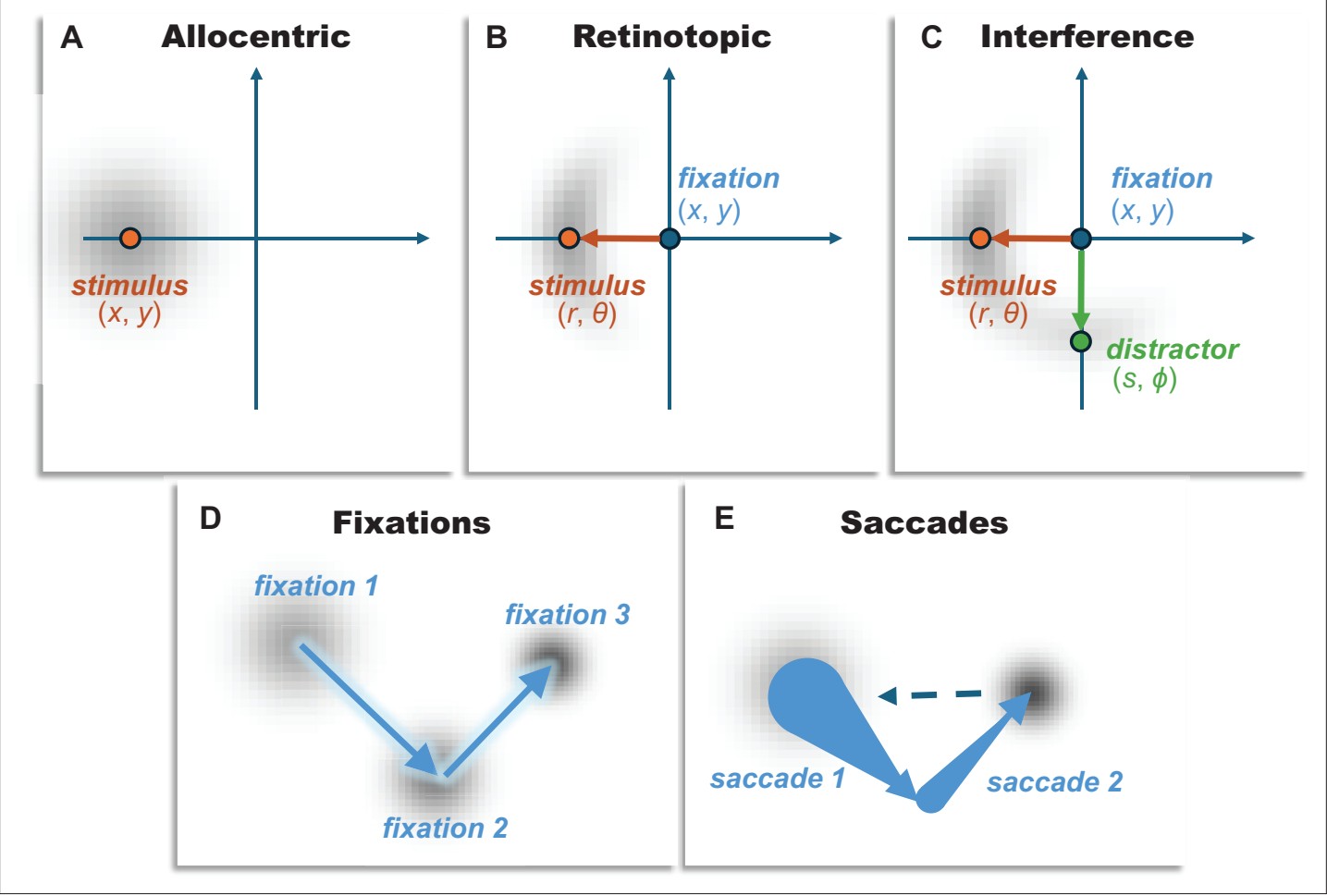

**Figure 3.** Conceptual illustration of allocentric vs. retinotopic spatial representations. This figure depicts the distinction between various representations of spatial memory. These include allocentric (world-centred) and retinotopic (eye-centred) spatial representation. (**A**) The allocentric panel illustrates a world-centred reference frame, where a stimulus location (x,y) is represented independently of eye position. The spread of the grey shaded area around the stimulus indicates for this representation the encoding error and subsequent decay since the stimulus was viewed. In this framework, memory error is assumed to increase over time due to a general decay of the stored coordinates, which is treated as a decay in Euclidean space, such that the probability density of a remembered location (coordinates (x,y)) is a normal distribution whose covariance grows with time centred on the stimulus. This model corresponds to Model 1 in **Table 1**. (**B**) The retinotopic panel presents an eye-centred reference frame, where the stimulus location is encoded relative to the current fixation point (x,y) using polar coordinates: radial distance (r) and angular direction (θ). The orange arrow indicates the vector from fixation to the stimulus, representing these encoded retinotopic coordinates. The grey shaded area here again represents the initial encoding error and subsequent decay, which can be decomposed into radial and angular components. This dual model, incorporating both encoding error and decay in radial and angular components, corresponds to Model 2 in **Table 1**. It forms the basis for more complex models (Models 3–7 in **Table 1**) that differentiate between decay of the reference frame itself (e.g. recall of fixation point) and updating based on saccade information. (**C**) The interference panel shows the combination of a stimulus and distractor (translated into the same retinotopic coordinates), with some probability associated with a response to the memory of the distractor stimulus. The lower two panels distinguish between whether the origin of the retinotopic plot is recalled based upon a remembered fixation location (**D**), or whether it is based upon a reconstruction after propagating back through remembered saccade vectors (**E**) —conceptually shown with arrows whose base is large to indicate the (cumulative) uncertainty in the origin points of each vector.

3. **Remapping source**. Within a dual-frame system, whether error arises from a passive decay of the remembered fixation point (**Figure 3D**), or from an active but noisy reconstruction based on the saccade vector itself (**Figure 3E**).

All models assume that memory for location decays over time, formalised as a diffusion (Brownian motion) process, where the precision of the remembered location decreases with increasing retention interval. The key differences between the models lie in what is decaying and when (see Materials and methods for mathematical details and detailed variance functions).

These conceptual distinctions give rise to a family of seven specific computational models (**Table 1**):

1. **Allocentric model**: This model assumes the brain stores locations in a single, stable, world-centred (allocentric) reference frame. In this model, memory error arises solely from a time-dependent decay of the stored coordinates, independent of any eye movements.

2. **Dual model**: This foundational model introduces the core hypothesis of a dual-reference frame. It posits that locations are initially encoded in eye-centred (retinotopic) coordinates (**radial** distance and **angular** direction from fixation), which are then subject to both initial encoding errors and time-dependent decay. This forms the basis for more complex models that explicitly distinguish between decay in the reference frame's centre and coordinates within that frame.

3. **Dual (fixation decay) model**: Building on the dual-frame hypothesis, this model proposes that to remap a location after a saccade, the brain uses its memory of the *original fixation point*. This memory trace of the eye's position itself decays over time, introducing a specific source of error into the remapped location. Of note, the time at which the memory of the fixation location starts to decay is the time of the first saccade following the disappearance of the stimulus. This means that, depending upon the experimental condition, the fixation memory starts to decay *after* the retinotopic memories start to decay.

4. **Dual (saccade update) model**: This model offers an alternative updating mechanism. Instead of relying on a memory of the absolute fixation location, it proposes that the brain uses a memory of the *saccade vector* itself (the direction and amplitude of the eye movement). The original location is then reconstructed based on this imperfect, noisy memory of the saccade, providing a different source of remapping error. Here, each saccade between the fixation at which the stimulus was shown and the response time injects an additional encoding error which has cumulative effects, in addition to the cumulative memory decay from decays in each of the saccade vector memories in the time since the saccade.

5. **Dual+interference model**: This model tests the influence of cognitive load by incorporating crosstalk between items in memory. It suggests that the memory for the target location is partially corrupted by the presence of the other, distracting item's location, adding a source of error independent of remapping. In effect, this means the response distribution is assumed to be a weighted sum between the distributions anticipated if responding to the target or to the distractor.

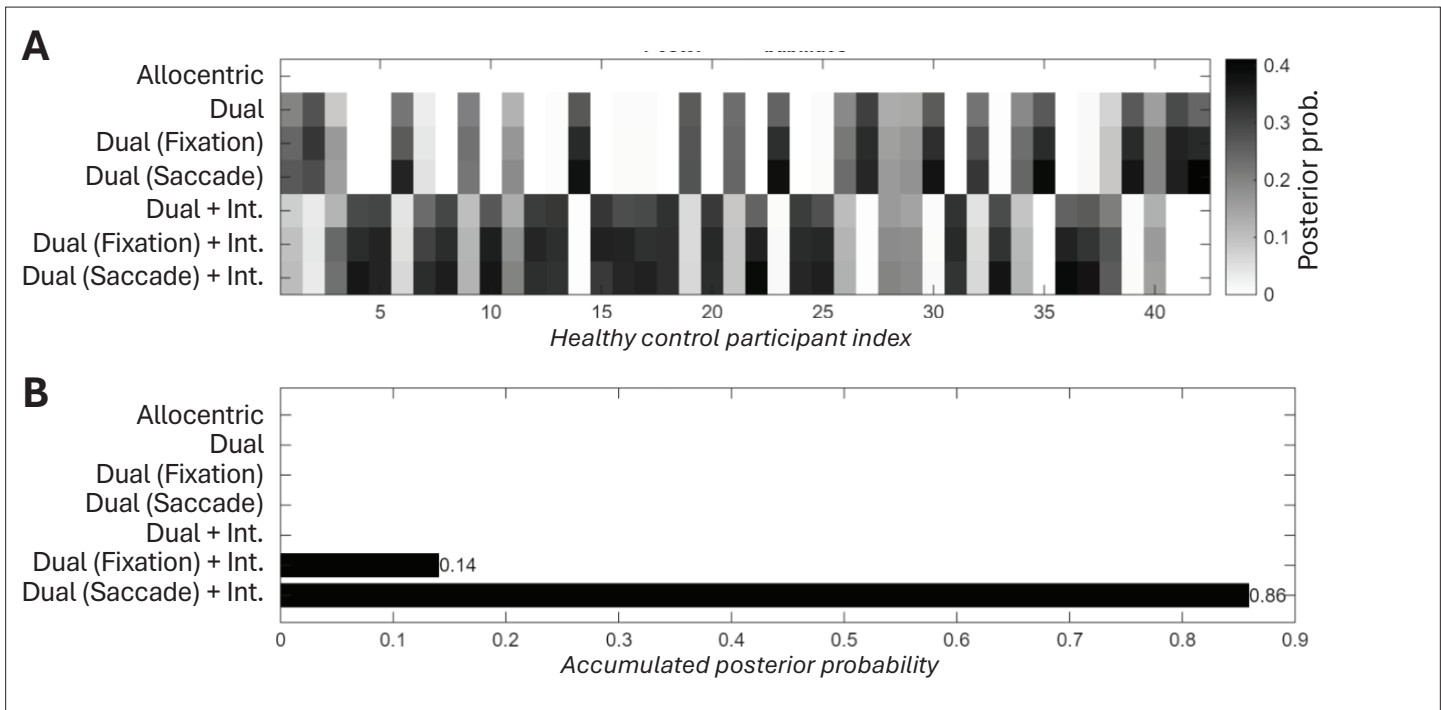

**Figure 4.** Model comparison results in healthy control (HC) participants (*N*=42). (**A**) Individual participant posterior probabilities. Heatmap illustrates the posterior probability for each of the seven competing models across individual healthy participants. This panel indicates that the 'Dual (Saccade Update)+Interference' model frequently shows the highest posterior probability for individual participants. (**B**) Aggregate model posterior probabilities. Bar chart showing the posterior probability for each competing model, aggregated across all healthy participants.

6. **Dual (fixation decay)+interference model**: This model combines two potential sources of error: the passive decay of the remembered fixation point (Fixation Decay) and corruption from the distractor item (Interference).
7. **Dual (saccade update)+interference model**: This is the most complex model, combining the active, saccade-based updating mechanism with distractor interference. It proposes that memory error arises from both an imperfect remapping process based on a noisy saccade signal and confusion with the other item held in memory.

By fitting these models to the trial-by-trial response data from all healthy participants (*N*=42), we adjudicated between competing mechanisms to determine which best explained participant performance (*Figure 4*). We used random-effects Bayesian model selection to identify the most plausible generative model. This process relies on the marginal likelihood (model evidence), which inherently balances model fit against complexity—a principle often referred to as Occam's razor (*Penny, 2012*; *Rasmussen and Ghahramani, 2000*; *MacKay, 2003*). The analysis yielded a strong result: the 'Dual (Saccade)+Interference' model (Model 7 in *Table 1*) emerged as the winning model, providing substantial evidence against the next best alternative with a Bayes factor of 6.11. This indicates substantial evidence that saccade-based updating with distractor interference is the most prevalent generative process in the healthy population.

The superiority of this specific model provides three critical insights. Firstly, the victory of a dual-frame model over the simpler allocentric model strongly supports the hypothesis that spatial memory is fundamentally reliant on eye-centred representations that must be actively updated. Secondly, the model comparison decisively adjudicates the source of remapping error: the clear win for the transsaccadic updating mechanism suggests that transsaccadic error arises not from a passive forgetting of where the eyes were, but from an active, yet imperfect, process of remapping based on a noisy internal copy of the saccade command—or, perhaps, from the proprioceptive signal associated with that movement. Finally, the inclusion of the 'Interference' term was essential for the winning model, demonstrating that this remapping process is also susceptible to confusion from other items held in memory. This winning model was therefore used for all subsequent analyses.

## Patients with AD and PD show preserved saccadic interference but greater overall error

Having established the mechanisms in healthy individuals, we next investigated how these processes are affected in patients with AD and PD, comparing their performance to that of both young (YC) and elderly (EC) controls. As anticipated, there was a significant main effect of group on overall memory performance. A two-way ANOVA revealed that patient groups were less accurate on both location ($F(3, 318)=59.71$, p<0.001) and colour memory ($F(3, 318)=47.85$, p<0.001). Post hoc analyses confirmed that AD patients exhibited significantly greater errors than all other groups across every condition (e.g. on No-Saccade Location Error vs. YC, EC, and PD: all p<0.001; see *Figure 5*).

Across all groups, saccades selectively disrupted spatial memory—shown by a significant main effect of condition on location error ($F(3, 318)=4.64$, p=0.003)—but not on colour error ($F(3, 318)=0.80$, p=0.49). Bayesian repeated-measures ANOVAs further supported this dissociation, providing moderate evidence for the null hypothesis in the AD group ($BF_{01}=3.35$, p(excl|data)=0.77) and weak evidence in the PD group ($BF_{01}=2.23$, p(excl|data)=0.69). This indicates that even in populations with established neurodegeneration, the detrimental impact of eye movements is specific to the spatial domain.

Most strikingly, the fundamental pattern of saccadic interference seen in healthy controls was fully replicated in the patient cohorts. In other words, the magnitude of this saccadic disruption did not differ between groups. We found no significant interaction between group and condition for either location ($F(9, 318)=0.46$, p=0.90) or colour memory ($F(9, 318)=0.26$, p=0.98). This critical null-interaction indicates that while overall spatial precision is lower in patients, the additional computational cost imposed by a saccade is not disproportionately larger.

The pattern of performance decay from the no-saccade to the two-saccade condition was remarkably consistent across all groups (*Figure 5A*), suggesting that the core mechanisms for updating spatial representations across eye movements are surprisingly resilient to the neurodegenerative processes in both PD and AD.

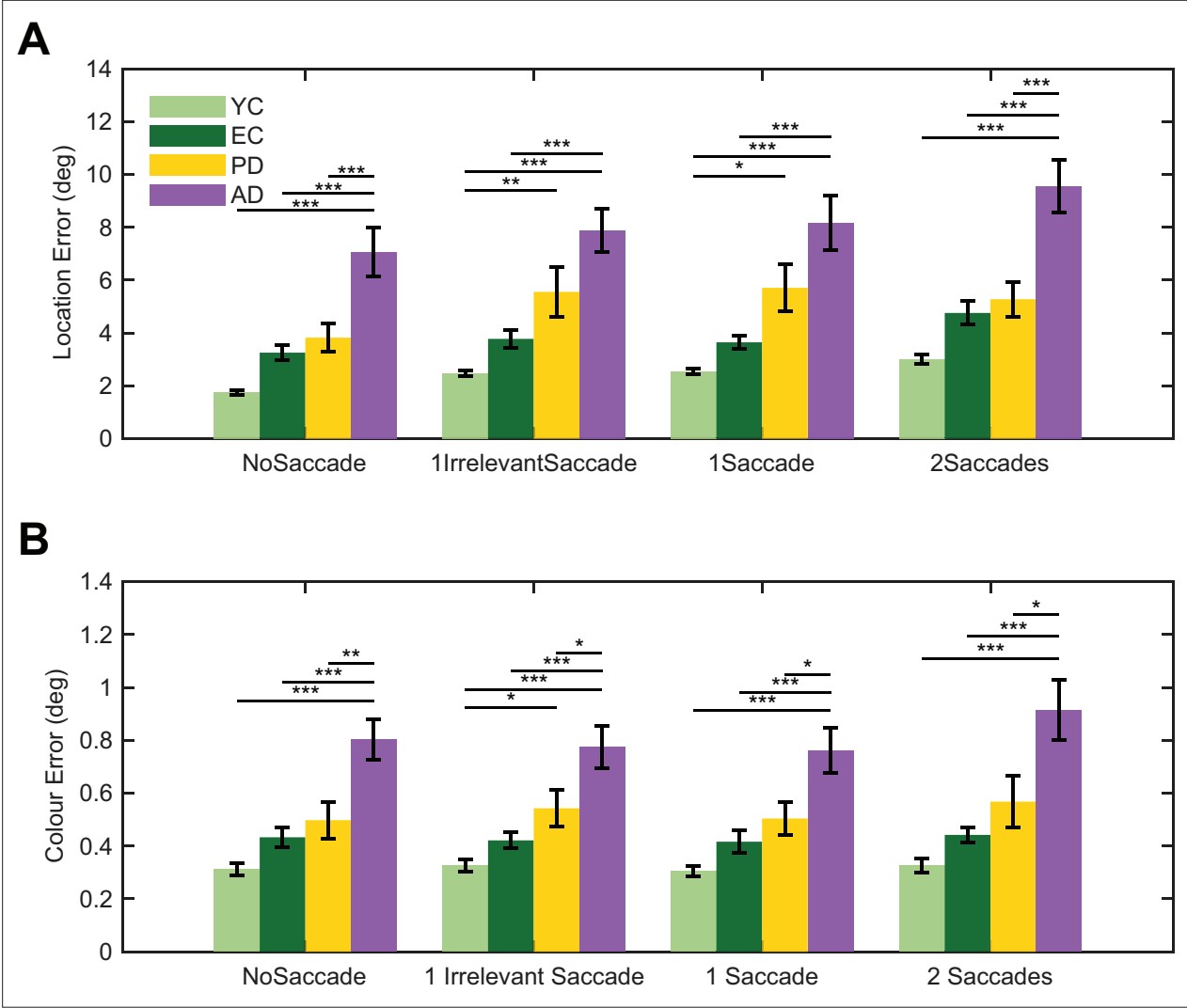

**Figure 5.** Saccadic interference with spatial memory is preserved across healthy ageing and neurodegenerative conditions, despite group differences in recall error. Saccades lead to increased recall error for location, but not colour, in healthy young (YC, *N* = 21) and elderly (EC, *N* = 21) controls, as well as in patients with Parkinson's disease (PD, *N* = 23) and Alzheimer's disease (AD, *N* = 22). While patient groups exhibit greater overall memory errors, the fundamental pattern of saccadic disruption remains consistent across all cohorts. (**A**) Location error. Mean location error (in degrees of visual angle) for each saccade condition (No-Saccade, 1 Irrelevant Saccade, 1 Saccade, 2 Saccades) for all participant groups. A two-way ANOVA revealed a significant main effect of Group and Condition, but there was no significant Group×Condition interaction ($F_{(9, 318)}=0.46$, $p=0.900$), indicating that the magnitude of saccadic disruption to spatial memory is constant across groups. Post hoc Tukey-Kramer comparisons within each condition between groups confirmed a graded impairment in overall spatial accuracy: AD patients exhibited significantly greater errors than all other groups (e.g. AD vs. YC, EC, PD: all $p<0.001$), PD patients performed worse than both YC and EC (all $p<0.001$), and EC performed worse than YC (all $p<0.001$). The significant post hoc between group comparisons are shown above the bar plots, with Bonferroni multiple comparison correction applied. (**B**) Colour error. Mean colour error (in radians) for each saccade condition across participant groups. For colour memory, a two-way ANOVA revealed a significant main effect of Group but no significant main effect of Condition or Group×Condition interaction. Post hoc comparisons for colour error showed a similar pattern of overall group differences as observed for location memory (all group comparisons $p<0.001$). Error bars represent ± 1 standard error of the mean (SEM). Asterisks denote statistical significance from post hoc comparisons: *: $p<0.05$, **: $p<0.01$; ***: $p<0.001$.

## Modelling reveals the sources of spatial memory deficits in healthy ageing and neurodegeneration

To understand the source of the observed deficits, we applied the winning 'Dual (Saccade)+Interference' model to the data from all participants (YC, EC, AD, and PD). By fitting the model to the entire dataset, we obtained estimates of the parameters for each individual, which then formed the basis for our group-level analysis. To formally test for group differences, we used parametric empirical

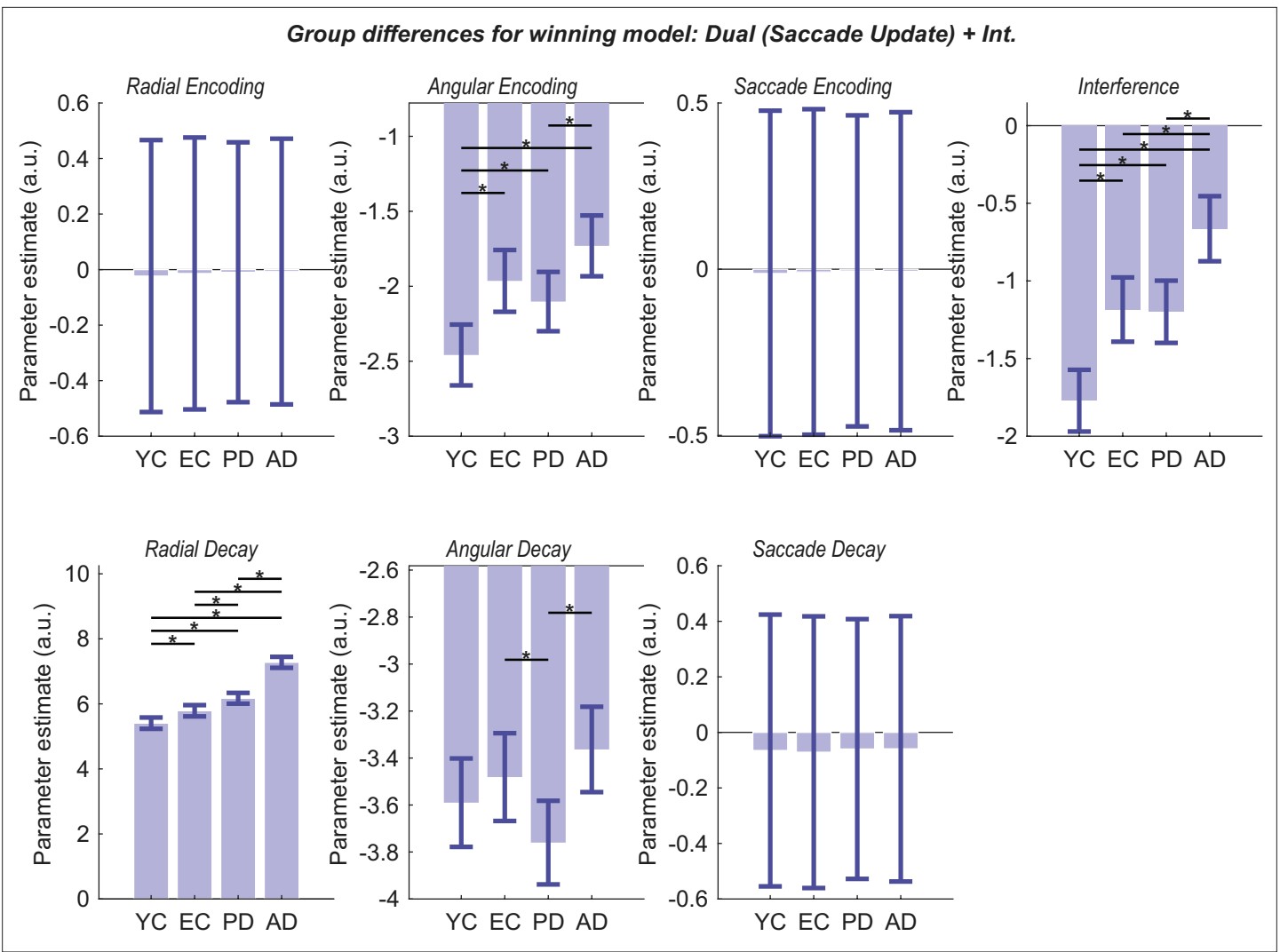

**Figure 6.** Group differences in winning model parameters. The plots show group-level parameter estimates from the parametric empirical Bayes (PEB) analysis for the winning 'Dual (Saccade)+Interference' model. The y-axis represents the posterior mean of the log-scaled parameter, where a more positive value indicates a greater influence of that parameter on performance (i.e. greater error or faster decay). The exception is the interference parameter, which is effectively a logit parameter, meaning the probability of selecting the distractor is obtained by a sigmoid transform. Error bars represent the 90% posterior confidence intervals. Group differences are indicated by an asterisk (*) if the posterior probability of a difference between the respective parameter estimates is greater than 95%, as calculated from their posterior distributions (i.e. if p(A>B)>0.95 or p(B>A)>0.95). The analysis reveals that Angular Encoding, Angular Decay, Radial Decay, and Interference parameters differentiate the groups. Critically, the EC group shows significant deviations from the YC group in Angular Encoding, Radial Decay, and Interference, providing a model-based profile of healthy age-related spatial memory decline. Notably, parameters related to saccade processing (Saccade Encoding and Saccade Decay) and Radial Encoding do not differ between groups, suggesting that the integrity of transsaccadic remapping is preserved across the lifespan and in early-stage neurodegeneration.

Bayes (PEB), a hierarchical Bayesian approach that compares parameter estimates across groups while accounting for the uncertainty of each estimate (*Zeidman et al., 2019*). This allowed us to identify which specific cognitive mechanisms, as formalised by the model parameters, were affected by age and disease.

The Bayesian inversion used here allows us to quantify the posterior mode and variance for each parameter and the covariance for each parameter. From these, we can compute the probabilities that pairs of parameters differ from one another, which we report as p(A>B)—meaning the posterior probability that the parameter for group A was greater than that for group B.

We first examined the specific parameters differentiating healthy elderly (EC) from young controls (YC) to isolate the factors contributing to non-pathological, age-related decline. The analysis revealed that healthy ageing is primarily characterised by a significant increase in Radial Decay

(p(EC>YC)=0.995), a heightened susceptibility to Interference (p(EC>YC)=1.000), and a reduction in initial Angular Encoding precision (p(YC<EC)=0.002; *Figure 6*). These results suggest that normal ageing degrades the fidelity of the initial memory trace and its resilience over time, while the core computational process of updating information across saccades remains intact.

Beyond these baseline ageing effects, our clinical cohorts exhibited more severe and condition-dependent impairments. Radial decay showed a clear, graded impairment: AD patients had a greater decay rate than PD patients (p(AD>PD)=1.000), who in turn were more impaired than the EC group (p(PD>EC)=0.996). A similar graded pattern was observed for Interference, where AD patients were most susceptible (p(AD>PD)=0.999), while the PD and EC groups did not significantly differ (p(PD>EC)=0.532).

Patients with AD also showed a tendency towards greater angular decay than controls (p(AD>EC)=0.772), although this fell below the 95% probability threshold. This effect was influenced by a lower decay rate in the PD group compared to the EC group (p(PD<EC)=0.037). In contrast, group differences in encoding were less pronounced. While YC exhibited significantly higher precision than all other groups, AD patients showed significantly higher angular encoding error than PD patients (p(AD>PD)=0.985), though neither group differed significantly from the EC group.

Crucially, parameters related to the saccade itself—saccade encoding and saccade decay—did not differentiate the groups. This indicates that neither healthy ageing nor the early stages of AD and PD significantly impair the fundamental machinery for transsaccadic remapping. Instead, the visuospatial deficits in these conditions arise from specific mechanistic failures: a faster decay of radial position information and increased susceptibility to interference, both of which are present in healthy ageing but significantly amplified by neurodegeneration.

## Transsaccadic working memory predicts clinical constructional deficits

To assess whether the mechanistic parameters derived from the LOCUS task represent core phenotypes of real-world visuospatial abilities, we also instructed all participants to complete the ROCF copy task (*Figure 7A*) on an Android tablet using a digital pen (see examples in *Figure 7B*; all Copy data are available in the open dataset: https://osf.io/95ecp/). The ROCF is a gold-standard neuropsychological tool for identifying constructional apraxia (*Zhang et al., 2021*). Historically, drawing performance has shown only weak or indirect correlations with traditional working memory measures (*Senese et al., 2020*). This disconnect has been attributed to active visual-sampling strategies—frequent eye movements that treat the environment as an external memory buffer, minimising the necessity of holding large volumes of information in internal working memory (*Draschkow et al., 2021*; *Ballard et al., 1995*; *Cohen, 2005*).

We hypothesised that drawing accuracy is primarily constrained by the precision of spatial updating across frequent saccades rather than raw memory capacity. To evaluate the ecological validity of the identified saccade-updating mechanism, we modelled individual ROCF copy scores across all four groups using the estimated (maximum a posteriori) parameters from the winning 'Dual (Saccade)+Interference' model (Model 7; *Figure 8*) as regressors in a Bayesian linear model. Prior to inclusion, each regressor was normalised by dividing by the square root of its variance.

This model successfully explained 61.99% of the variance in ROCF copy scores, indicating that these computational parameters are strong predictors of real-world constructional ability (*Figure 8A*). The posterior covariance matrix, illustrating the regression coefficients associated with each of the model parameters, is shown in *Figure 8B*. *Figure 8C* indicates how specific mechanistic parameters during transsaccadic working memory contributed to explaining ROCF copy performance. Although, as shown in *Figure 6*, the saccade encoding error did not differentiate across these four groups, it does contribute to a worse ROCF copying performance. This highlights the critical role of accurate remapping based on saccadic information; even if the core saccadic update mechanism is preserved across groups (as shown in previous analyses), the precision of this updating process is crucial for complex visuospatial tasks. Moreover, worse ROCF copy performance is associated particularly with higher initial angular encoding error. This indicates that imprecision in the initial registration of angular spatial information contributes to difficulties in accurately reproducing complex visual stimuli.

Surprisingly, less angular decay was associated with poorer ROCF copy scores. However, a plausible explanation for this could be related to the trend seen in angular encoding error. People who have smaller angular encoding errors have more room for memory decay before angular encoding

**A**

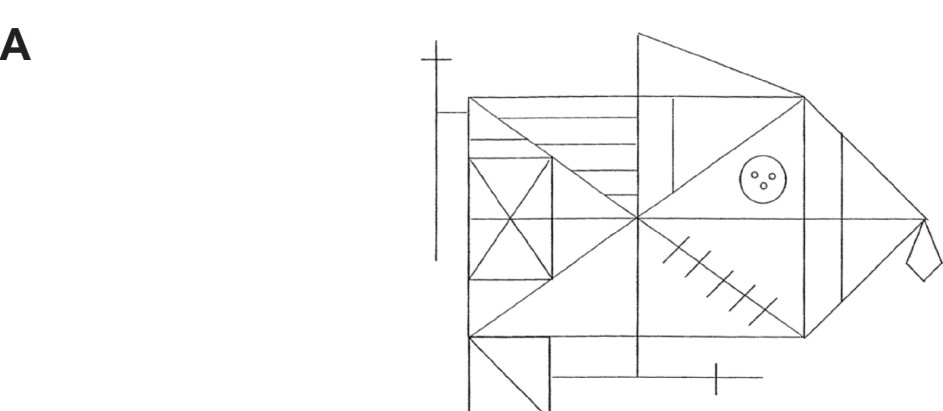

**B** *Example 1 (Age = 28, ACE Total score= 97)*

Copy | Immediate Recall

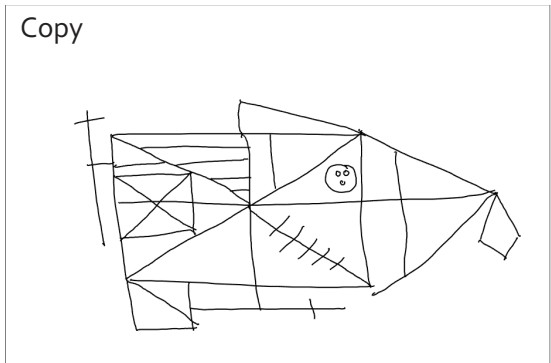
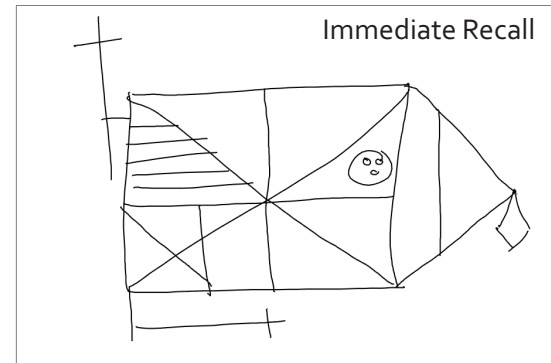

*Example 2 (Age = 75, ACE Total score= 95)*

Copy | Immediate Recall

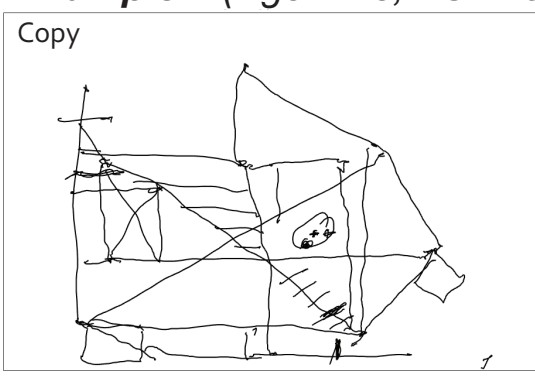
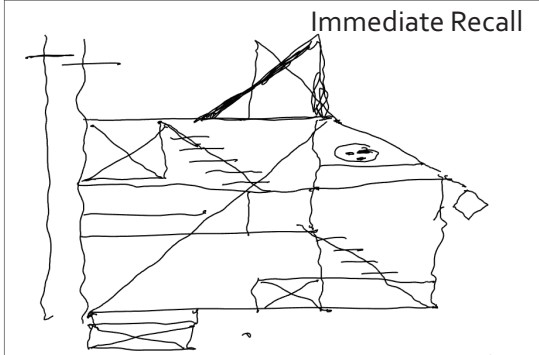

**Figure 7.** Rey-Osterrieth complex figure (ROCF) copying performance. (**A**) The original figure that participants copied and recalled. (**B**) Examples of ROCF copy and immediate recall tasks from a young and an elderly healthy participant, both with normal cognitive function (ACE total score>88). ACE = Addenbrooke's Cognitive Examination-III.

errors accumulate. Evidence for this explanation is evident in the (weak) negative correlation evident in the covariance matrix in *Figure 8* between the coefficients for angular encoding and decay rates. The implication is that the results would be similar if we slightly increased the encoding variance while reducing the decay rate, and vice versa.

Conversely, parameters such as radial encoding error, radial decay, saccade decay and interference did not predict ROCF copy scores. This differential predictive power underscores the specific relevance of angular and saccade-related precision for the demanding task of copying complex figures, which inherently requires continuous and accurate spatial updating across multiple fixations.

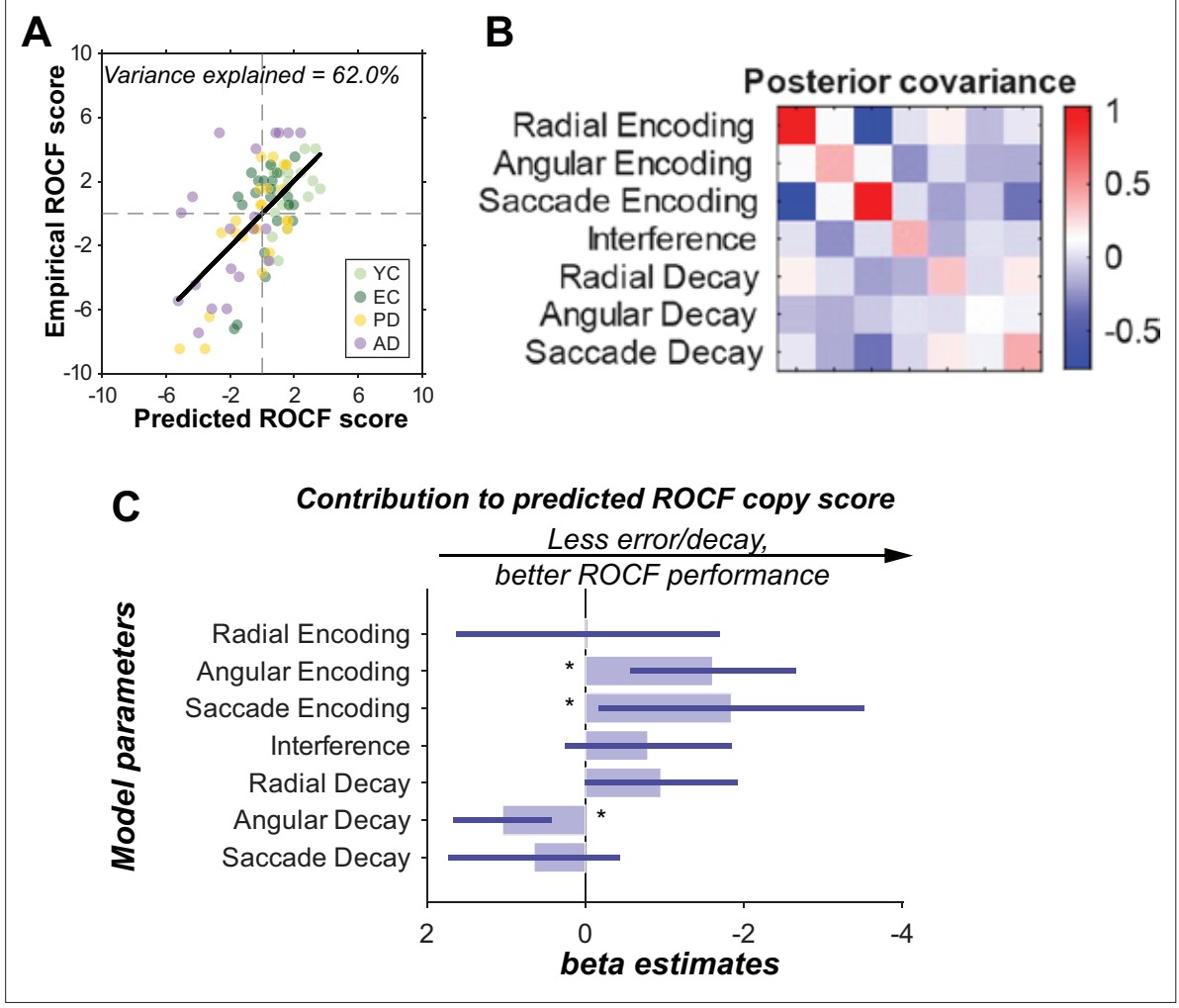

**Figure 8.** Effect of model parameters on Rey-Osterrieth complex figure (ROCF) copy scores. How model parameters from the winning 'Dual (Saccade)+Interference' model explain variance in ROCF copy performance. (**A**) Predicted vs. empirical scores. A scatter plot of empirical ROCF scores against scores predicted by the fitted linear model. Each point represents an individual participant, coloured by diagnostic group: young controls (YC; light green), elderly controls (EC; dark green), Parkinson's disease (PD; yellow), and Alzheimer's disease (AD; purple). A black line denotes the linear fit across all data. ROCF scores were demeaned across participants. Scores outside of 2 standard deviations from the mean were omitted. The plot title indicates the percentage of variance in ROCF copy scores explained by the model. (**B**) Posterior covariance matrix. A heatmap depicting the posterior covariance between the estimated model parameters (i.e. between the regression coefficients that determine the contribution of each of the parameters from our previous winning model to the ROCF score). Diagonal elements represent individual parameter variances; off-diagonal elements show covariances between parameter pairs. The colour bar provides the scale. (**C**) Estimated parameter contributions. Bar plot displaying the estimated contribution (beta coefficients) of each LOCUS model parameter to the predicted ROCF copy score. Error bars represent 90% credible intervals (1.65 standard deviations of the posterior variance). Asterisks (∗) denote parameters whose 90% confidence interval does not encompass zero.

Furthermore, to evaluate the specificity of the relationship between transsaccadic updating and constructional ability, we conducted a partial correlation analysis. The association between the trans-saccadic working memory in LOCUS and ROCF copy performance remained highly significant (rho = –0.46, p<0.001), even after accounting for age, education, and general memory performance (ACE-III Memory subscore).

## Discussion

This study introduced a novel paradigm, the LOCUS task (*Figure 1*), and combined this with computational modelling to investigate the underlying mechanisms and sources of potential errors in dynamic spatial working memory, particularly how the brain maintains a stable world-centred frame of reference

across saccades. We investigated this across four cohorts: young healthy adults (YC), healthy elderly adults (EC), and patients with AD and PD. The two clinical groups are known to be frequently associated with significant disruptions to visuospatial processing and oculomotor control (*Cormack et al., 2004*; *Trojano and Gainotti, 2016*).

The LOCUS task was designed to dissociate retinotopic (within-fixation) and transsaccadic (across-fixation) working memory, examining the precision of both location and colour information maintained across saccades. Our findings reveal a 'saccade cost'—a reduction in spatial memory precision following eye movements, consistent with previous research on healthy young adults (*Golomb and Kanwisher, 2012*). Unexpectedly, this saccade cost was remarkably consistent across all four populations, demonstrating its stability with ageing and neurodegeneration. This finding implies that any observed reduction in transsaccadic performance in older adults and neurodegenerative patients is primarily attributable to lower baseline retinotopic memory rather than an increased cost associated with saccadic remapping.

The conclusion was further supported by our computational modelling. The winning model, 'Dual (Saccade)+Interference', posits an eye-centred, retinotopic representation (encoding stimulus as angular and radial displacement from fixation) that also incorporates saccade-based updating and interference. Although saccade updating was an essential component of the winning model, its two key parameters—initial encoding error and decay rate during maintenance—did not significantly differ across groups. This indicates that the core computational process of updating spatial information based on eye movements is largely preserved in healthy ageing and neurodegeneration.

Instead, group differences were driven by deficits in angular encoding error (precision of initial angle from fixation), angular decay, radial decay (decay in memory of distance from fixation), and interference susceptibility. This implies a functional and neuroanatomical dissociation: while the ventral stream (the 'what' pathway) shows an age-related decline in the quality and stability of stored representations, the dorsal stream (the 'where' pathway) parietal-frontal circuits responsible for coordinate transformations remain functionally robust (*Ungerleider and Mishkin, 1982*; *Mishkin et al., 1983*; *Cohen and Andersen, 2002*; *Land, 2012*). These spatial updating mechanisms appear resilient to the normal ageing trajectory and only break down when challenged by the specific pathological processes seen in AD or PD.

The eye-centred retinotopic representation identified by our model is consistent with findings in posterior parietal cortex neurons (*Cohen and Andersen, 2002*; *Land, 2012*), which are considered to play a crucial role in visual localisation, as well as remapping across eye movements (*Duhamel et al., 1992*; *Ten Brink et al., 2019*). In using this computational approach, we align our work with established practices in computational psychiatry: Our framework employs Variational Laplace, a method used to recover computational phenotypes in clinical populations like those with substance use disorders (*Smith et al., 2023*; *Taylor et al., 2023*), and the models we fit using this procedure feature time-dependent parameterisation of variance—conceptually similar to the widely used Hierarchical Gaussian Filter (*Mathys et al., 2014*; *Hess et al., 2025*; *Marshall et al., 2016*; *Kirsch et al., 2021*). While we describe the present framework as 'computational', it is more precisely characterised as an algorithmic-level generative model within Marr's hierarchy. Our focus was on defining the rules of spatial integration and the sources of eye-movement-induced noise, rather than deriving these processes from normative principles or defining their specific neural implementation. Importantly, the risk of overfitting is mitigated by the Bayesian model selection framework; by utilising the marginal likelihood for model comparison, the procedure inherently penalises excessive model complexity and promotes generalisability (*Penny, 2012*; *Rasmussen and Ghahramani, 2000*; *MacKay, 2003*; *Murray and Ghahramani, 2005*). This generalisability was further evidenced by the model's ability to predict performance on the independent ROCF task, confirming that these parameters represent robust mechanistic phenotypes rather than idiosyncratic fits to the initial dataset.

A clear finding was the specificity of the saccade cost to spatial features; it was not observed for non-spatial features like colour, even in neurodegenerative conditions. This discrepancy challenges notions of fixed visual working memory capacity unaffected by saccades (*Irwin, 1992*; *Prime et al., 2011*; *Prime et al., 2007*; *Irwin and Gordon, 1998*). The differential impact on spatial vs. non-spatial features in transsaccadic memory aligns with the established 'what' and 'where' pathways in visual processing (*Ungerleider and Mishkin, 1982*; *Mishkin et al., 1983*). For objects to remain unified, object features must be bound to stable representations of location across saccades (*Cavanagh et al.,*

*2010*). One possibility is that remapping updates both features and location through a shared mechanism, predicting equal saccadic interference for both colour and location in the present study.

However, our findings suggest otherwise. One potential concern is whether this dissociation simply reflects the inherent spatial noise introduced by fixational eye movements (FEMs), such as microsaccades and drifts (*Engbert and Kliegl, 2004*). Because locations are stored in a retinotopic frame, fixational instability necessarily shifts retinal coordinates over time. However, the 'saccade cost' here was defined as the error increase relative to a no-saccade baseline of equal duration; because both conditions are subject to the same fixational drift, any FEM-induced noise is effectively subtracted out. Thus, despite the ballistic and non-Gaussian nature of FEMs (*Mergenthaler and Engbert, 2007*), they cannot account for the saccade cost in spatial memory and its total absence in the colour domain. Another possibility is that this dissociation reflects differences in baseline task difficulty or dynamic range. Yet, the presence of a robust recency effect in colour memory (*Figure 2B*) confirms that our paradigm was sensitive to memory-dependent variance and was not limited by floor or ceiling effects.

The fact that identical eye movements—executed simultaneously and with identical vectors—systematically degraded spatial precision while sparing colour suggests a feature-specific susceptibility to transsaccadic remapping. This supports the view that the computational process of updating an object's location involves a vector subtraction mechanism—incorporating noisy oculomotor commands (efference copies)—that introduces specific spatial variance. Because this remapping is a coordinate transformation, the resulting sensorimotor noise does not functionally propagate to non-spatial feature representations. Consequently, features like colour may be preserved or automatically remapped without the precision loss associated with spatial updating (*Golomb and Mazer, 2021*; *Fabius et al., 2020*). Our paradigm thus provides a refined tool to investigate the architecture of transsaccadic working memory across distinct object features.

Importantly, our computational framework establishes a direct mechanistic link between transsaccadic updating and real-world constructional ability. Specifically, higher saccade and angular encoding errors contribute to poorer ROCF copy scores. By mapping these mechanistic estimates onto clinical scores, we found that the parameters derived from our winning model explain approximately 62% of the variance in constructional performance across groups. These findings suggest that the computational parameters identified in the LOCUS task represent core phenotypes of visuospatial ability, providing a mechanistic bridge between basic cognitive theory and clinical presentation.

This relationship provides novel insights into the cognitive processes underlying drawing, specifically highlighting the role of transsaccadic working memory. Previous research has primarily focused on the roles of fine motor control and eye-hand coordination in this skill (*Ballard et al., 1995*; *Cohen et al., 2021*; *Raimo et al., 2021*; *Trojano et al., 2009*; *Bai et al., 2021*; *Gowen and Miall, 2006*; *Petilli et al., 2021*). This is partly because of consistent failure to find a strong relation between traditional memory measures and copying ability (*Ballard et al., 1995*; *Senese et al., 2020*). For instance, common measures of working memory, such as digit span and Corsi block tasks, do not directly predict ROCF copying performance (*Senese et al., 2020*; *Kim et al., 2022*). Furthermore, in patients with constructional apraxia, these memory performance measures often remain relatively preserved despite significant drawing impairments (*Kim et al., 2022*; *Papagno, 2002*; *Russell et al., 2010*). In the literature, this lack of association has often been attributed to 'deictic' visual-sampling strategies, characterised by frequent eye movements that treat the environment as an external memory buffer, thereby minimising the need to maintain a detailed internal representation (*Ballard et al., 1995*; *Tchalenko and Chris Miall, 2009*). In a real-world copying task, the ROCF requires a high volume of saccades, making it uniquely sensitive to the precision of the dynamic remapping signals identified here. Recent eye-tracking evidence confirms that patients with AD exhibit significantly more saccades and longer fixations during figure copying compared to controls, potentially as a compensatory response to transsaccadic working memory constraints (*Kim et al., 2022*). This high-frequency sampling—averaging between 150 and 260 saccades for AD patients compared to approximately 100 for healthy controls—renders the task highly dependent on the precision of dynamic remapping signals (*Kim et al., 2022*). To ensure this relationship was not driven by a general '*g*-factor' or non-spatial memory impairment, we further investigated the role of broader cognitive performance using the ACE-III Memory subscale. We found that the relationship between transsaccadic working memory and ROCF performance remains highly significant, even after controlling for age, education, and ACE-III Memory subscore. This suggests that transsaccadic updating may represent a discrete

computational phenotype required for visuomotor control, rather than a non-specific proxy for global cognitive decline (*Russell et al., 2010*).

In other words, even when visual information is readily available in the world, the act of copying depends critically on working memory across saccades. This reveals a fundamental computational trade-off: while active sampling strategies (characterised by frequent eye-hand movements) effectively reduce the load on capacity-limited working memory, they simultaneously increase the demand for precise spatial updating across eye movements. By treating the external world as an 'outside' memory buffer, the brain minimises the volume of information it must hold internally, but it becomes entirely dependent on the reliability with which that information is remapped after each eye movement. This perspective aligns with, rather than contradicts, the traditional view of active sampling, which posits that individuals adapt their gaze and memory strategies based on specific task demands (*Draschkow et al., 2021*; *Droll and Hayhoe, 2007*). Furthermore, this perspective provides a mechanistic framework for understanding constructional apraxia; in these clinical populations, the impairment may not lie in a reduced memory 'span', but rather in the cumulative noise introduced by the constant spatial remapping required during the copying process (*Russell et al., 2010*; *Van der Stigchel et al., 2018*).

Beyond constructional ability, these findings suggest that the primary evolutionary utility of high-resolution spatial remapping lies in the service of action rather than perception. While spatial remapping is often invoked to explain perceptual stability (*Golomb and Mazer, 2021*; *Melcher and Colby, 2008*; *Bays and Husain, 2007*; *Harrison et al., 2024*), the necessity of high-resolution transsaccadic memory for basic visual perception is debated (*Bays and Husain, 2007*; *Irwin et al., 1983*; *Irwin et al., 1988*; *Rayner et al., 1980*). A prevailing view suggests that detailed internal models are unnecessary for perception, given the continuous availability of visual information in the external world (*Bays and Husain, 2007*; *Prime et al., 2011*). Our findings support an alternative perspective, aligning with the proposal that high-resolution transsaccadic memory primarily serves action rather than perception (*Bays and Husain, 2007*). This is consistent with the need for precise localisation in eye-hand coordination tasks such as pointing or grasping (*Baltaretu et al., 2020*). Even when unaware of intrasaccadic target displacements, individuals rapidly adjust their reaching movements, suggesting direct access of the motor system to remapping signals (*Prablanc and Martin, 1992*). Further support comes from evidence that pointing to remembered locations is biased by changes in eye position (*Henriques et al., 1998*), and that remapping neurons reside within the dorsal 'action' visual pathway, rather than the ventral 'perception' visual pathway (*Bays and Husain, 2007*; *Husain and Nachev, 2007*; *Goodale and Westwood, 2004*). By demonstrating a strong link between transsaccadic working memory and drawing (a complex fine motor skill), our findings suggest that precise visual working memory across eye movements plays an important role in complex fine motor control.

## Materials and methods
### Participants

A total of 87 participants completed the study: 21 young healthy adults (YC), 21 older healthy adults (EC), 23 patients with PD, and 22 patients with AD. Their demographic and clinical details are summarised in *Table 2*. Initially, 90 participants were recruited (22 YC, 21 EC, 25 PD, 22 AD); however,

**Table 2.** Demographics of participants.
The value in brackets indicates 1 standard deviation or proportion of the sample. YC = young healthy control; EC = elderly healthy control; PD = Parkinson's disease; AD = Alzheimer's disease; ACE = the Addenbrooke's Cognitive Examination III. Two relevant ACE-III subscores are also presented (memory and visuospatial).

| Group | N | Age | Gender (*N* of females) | Handedness (*N* of right handed) | Education (years) | ACE total | ACE memory | ACE visuospatial |
|---|---|---|---|---|---|---|---|---|
| YC | 21 | 24.1 (4.7) | 15 (71.4%) | 18 (85.7%) | 17.0 (2.0) | 96.6 (2.9) | 25.0 (1.2) | 15.7 (0.6) |
| EC | 21 | 72.4 (6.3) | 12 (57.1%) | 18 (85.7%) | 16.6 (4.9) | 96.1 (3.5) | 24.0 (2.9) | 15.8 (0.7) |
| PD | 23 | 70.8 (6.2) | 11 (47.8%) | 21 (91.3%) | 15.7 (3.4) | 93.7 (4.9) | 24.2 (2.6) | 15.3 (0.9) |
| AD | 22 | 68.2 (9.7) | 9 (40.9%) | 16 (72.7%) | 14.6 (4.1) | 78.3 (13.0) | 17.8 (5.3) | 13.5 (2.2) |

three individuals (1 YC and 2 PD) were excluded from all analyses due to technical issues during data acquisition.

All participants were recruited locally in Oxford, UK. None were professional artists, had a history of psychiatric illness, or were taking psychoactive medications (excluding standard dopamine replacement therapy for PD patients). Young participants were recruited via the University of Oxford Department of Experimental Psychology recruitment system. Older healthy volunteers (all >50 years of age) were recruited from the Oxford Dementia and Ageing Research (OxDARE) database.

Patients with PD were recruited from specialist clinics in Oxfordshire. All had a clinical diagnosis of idiopathic PD and no history of other major neurological or psychiatric conditions. While specific dosages of dopamine replacement therapy (e.g. levodopa equivalent doses) were not systematically recorded, all patients were tested while on their regular medication regimen ('ON' state).

Patients with PD were recruited from clinics in the Oxfordshire area. All had a clinical diagnosis of idiopathic PD and no history of other major neurological or psychiatric illnesses. While all patients were tested in their regular medication 'ON' state, the specific pharmacological profiles—including the exact types of medication (e.g. levodopa, dopamine agonists, or combinations) and dosages—were not systematically recorded. The disease duration and PD severity were also unrecorded for this study.

Patients with AD were recruited from the Cognitive Disorders Clinic at the John Radcliffe Hospital, Oxford, UK. All AD participants presented with a progressive, multidomain, predominantly amnestic cognitive impairment. Clinical diagnoses were supported by structural MRI and FDG-PET imaging consistent with a clinical diagnosis of AD dementia (e.g. temporo-parietal atrophy and hypometabolism) (*McKhann et al., 2011*). All neuroimaging was reviewed independently by two senior neurologists (ST and MH).

Global cognitive function was assessed using the Addenbrooke's Cognitive Examination-III (ACE-III) (*Mathuranath et al., 2000*). All healthy participants scored above the standard cut-off of 88, with the exception of one elderly participant who scored 85. In the PD group, two participants scored below the cut-off (85 and 79). In the AD group, six participants scored above 88; these individuals were included based on robust clinical and radiological evidence of AD pathology rather than their ACE-III score alone.

## Experimental procedure

The LOCUS task (*Figure 1*) assessed visuospatial working memory across eye movements. Participants viewed two sequentially presented coloured squares (1 s duration each, with a 0.5 s inter-stimulus interval), memorising their location and colour within frames centred on a fixation cross. A labelled colour wheel then cued recall of a specific square. Participants selected the remembered colour and dragged a corresponding square to its original screen location.

Saccade demands were manipulated by varying the location of the second frame and colour wheel. In 50% of trials, the second frame appeared in the same location as the first (no saccade required), while in the other 50%, it appeared in a different location, requiring a saccade of at least 8.5° (degrees of visual angle). The colour wheel also appeared in the same or a different location as the second frame with equal probability. This creates four conditions: No-saccade, Saccade-After-Item 1, Saccade-After-Item-2, and Two-Saccades (*Figure 1*). Each condition comprised 25% of the 160 trials, presented in 16 randomised blocks.

The coloured square was 0.66°×0.66° visual angle, the fixation cross was 1°×1° visual angle, and the frame was 9.8°×9.8° visual angle with a 0.1° linewidth. The location of the square was randomised within the frame with a fixed minimal distance from fixation cross of 3.82° visual angle. The screen had 3×5 possible frame locations with a 0.3° gap between frames. The colour wheel was 4.5° wide, with a 1.5°-wide inner black circle displaying a 1° tall label (probing '1' or '2'). The colour and location of squares were generated pseudorandomly for each participant just before the starting experiment. To avoid the potential bias to certain locations on the colour wheel, the colour wheel's rotation was randomised for every trial.

Participants were instructed to maintain fixation on a cross when it was present and to execute saccades to peripheral targets when prompted by appearance of a new frame at a different location or the appearance of the colour wheel. When Item 1 or 2 was displayed, it was positioned within a square frame with a central fixation cross (*Figure 2*). The use of a square frame and central fixation

cross served to emphasise the retinotopic reference frame, highlighting no need to move the eyes unless these elements were extinguished, and a new stimulus appeared in the periphery. Thus, in the no-saccade condition, participants did not need to make eye movements and therefore respond without having to rely on transsaccadic memory.

To mitigate potential confounds related to visual impairment or inability to fixate with task instructions, we monitored eye position throughout the experiment. Eye-tracking analysis confirmed high compliance: participants fixated the cross in the majority of trials (81% ± 10%), with no significant difference between age groups ($t(40) = 0.76$, p=0.45). Fixation durations averaged 0.21 s (±0.05 s), sufficient for encoding visual information. Furthermore, in the no-saccade condition, participants successfully maintained fixation in 80% of trials (±10%). Similarly, in the saccade conditions, participants accurately executed saccades away when required on 86% of trials (±18%). Non-compliance trials were excluded from further analysis.

Recall was performed in a fixed order, with colour reported before location. This sequence was primarily chosen to minimise cognitive load and task-switching demands for the two neurological patient cohorts, ensuring the paradigm remained accessible for individuals with AD and PD. While this order results in a slightly longer retention interval for location recall, the saccade cost was identified by comparing location error across experimental conditions with similar timings but varying saccadic demands.

## Measurement of saccade cost

The LOCUS task allows for the quantification of several key metrics: (1) retinotopic working memory: the location error in no-saccade condition, (2) transsaccadic working memory: the location error in two-saccades condition, and (3) 'saccade cost': the reduction in spatial memory accuracy attributable to saccades.

Within this framework, it is important to distinguish between the broad construct of spatial working memory and the specific process of transsaccadic memory. While spatial working memory refers to the general ability to maintain spatial information over short intervals, transsaccadic memory describes the dynamic updating of these representations—termed remapping—to ensure stability across eye movements. Unlike classical 'static' measures of spatial working memory, such as the Corsi block task which focuses on memory span, transsaccadic memory tasks explicitly require the integration of stored visual information with motor signals from intervening saccades. Our paradigm treats transsaccadic updating as a core computational process within spatial working memory, where eye-centred representations are actively reconstructed based on noisy memories of the intervening saccade vectors.

While a straightforward approach to quantify saccade cost would be to directly compare performance in the two-saccades condition to the no-saccade condition, this method confounds saccade cost with retinotopic working memory. To isolate the effect of saccades on memory independent of retinotopic condition, we calculated saccade cost as the difference in performance between the Two-Saccade condition and the Saccade-After-The-Other-Item condition. This yields a specific measure of the additional memory error due to a saccade made immediately after the probed item.

Furthermore, we conducted a secondary analysis using the beta coefficient derived from a multiple linear regression model to estimate the effect of saccades and item order on location errors. This approach confirmed the relationship between saccade cost and ROCF copy performance, demonstrating the robustness of our findings.

## Rey-Osterrieth complex figure

Participants completed the ROCF copy and immediate recall tasks on a Galaxy Tab S3 tablet (SM-T820) using the corresponding digital pen (Galaxy Tab S3 S Pen). The tablet's screen dimensions were 148×197 mm² (1536×2048 pixels, 264 ppi density). Participants placed the tablet on the table upright. During the copy phase, the ROCF figure displayed on the top of the tablet (*Rey, 1941*). They first copied the figure in the lower portion of the screen. Upon completion, they were instructed to reproduce the ROCF from memory, without the figure present. Drawing traces were recorded and printed for blind scoring by two experienced raters using the standard 18-element scoring system (*Osterrieth, 1944*).

For further analysis, the drawing paths were analysed in MATLAB using in-house scripts. We extracted the total time taken to complete each task, calculated from the first moment the digital

pen touched the screen to the final pen lift. In addition, we extracted total distance drawn (total path length in pixels), path drawing speed (mean derivative of path distance per second), and mean waiting time (duration between completing the last path and starting the next path).

## Computational modelling

We used a series of simple computational models to test a set of hypotheses about the mechanisms that underwrite the measured data. All hypotheses rely upon the idea that memory decays with time since the stimulus was presented, modelled here using a diffusion (or Brownian motion) like process. Under this formulation, the shape of the distribution of expected responses, and its change over time, will depend upon the way in which our brains represent the variables that we expect to decay. Conceptually, our hypotheses distinguished between allocentric (screen centred) coordinates and retinotopic coordinates, whether target and distractor representations do or do not interfere with one another, whether there was evidence for an embodied representation in which the current position of the eyes helps reinforce a memory, and, if so, whether remembered locations are centred on remembered fixations or on relative position based upon memories of the vectors of saccades performed since that location.

*Figure 9* shows the form of these hypotheses, each of which can be expressed with a time-dependent likelihood function. Each of these models was fit to the responses made by each participant, where the response data were rotated and translated such that the target stimulus was always in the same angle, and the origin was always set to be the fixation location associated with the probed stimulus. The model fits were performed using a Variational Laplace procedure (a form of approximate Bayesian inference) using a Newton optimisation scheme as implemented in SPM25 using the spm_nlsi_Newton.m MATLAB routine (*Tierney et al., 2025*; *Zeidman et al., 2023*). This relies upon a specification of a log likelihood function, which gives the distribution, given the parameters, of the recorded responses. In the allocentric model, this log likelihood function had the form:

$$L\left(\kappa\right) = ln\left(\mathcal{N}\left(u; x, \kappa_1\tau + \kappa_2\right) \times \mathcal{N}\left(v; y, \kappa_1\tau + \kappa_2\right)\right)$$

Here, $(u,v)$ are the Euclidean coordinates of the response to a stimulus presented at coordinates $(x,y)$, with an encoding variance of $\kappa_2$ and a decay constant of $\kappa_1$ which scales with the time $\tau$ between the stimulus presentation and the response. This is summed over all responses a participant has made, accounting for the time at which they are made to give the accumulated log likelihood for the parameters for that dataset.

The dual models, with a retinotopic component, all have the form:

$$L\left(\kappa\right) = ln\left(\mathcal{N}\left(\sqrt{\left(u - x\right)^2 + \left(v - y\right)^2}; r, \Sigma_r\left(\kappa, \tau, t, T\right)\right) \times \mathcal{N}\left(\tan^{-1}\left(\frac{v - y}{u - x}\right); \theta, \Sigma_\theta\left(\kappa, \tau, t, T\right)\right)\right)$$

Here, the principle is similar, but the distributions are over the polar coordinates of the response, with $(r,\theta)$ representing the polar coordinates of the stimulus. The variances, $\Sigma$, are now functions of the parameter set and (up to) three different time variables. These different times are illustrated schematically in *Figure 9* (middle panel) and are disambiguated by the different experimental conditions. They are:

1. the time since the stimulus was visible ($\tau$)
2. the time since the first saccade (if any) since the stimulus was present ($t$)
3. the time since the second saccade (if any) after the stimulus ($T$).

The forms of the variance functions are shown in *Figure 9*. Here, we see a combination of encoding parameters (time-independent), and decay parameters for retinotopic stimulus memories (multiplied by $\tau$), fixation locations (multiplied by $t$), and saccade vectors (multiplied by $t+T$). The reason for multiplying saccade decay constants by the sum of $t$ and $T$ is that a reconstructed fixation location based upon saccade vectors will have a cumulative increase in uncertainty with each saccade since that fixation.

Finally, the likelihood for the interference models is the weighted sum of two likelihood ($e^{L\left(\kappa\right)}$) functions of the form above, but with one computed from the target location and the other from the distractor location. The interference parameter determined the weighting towards the distractor.

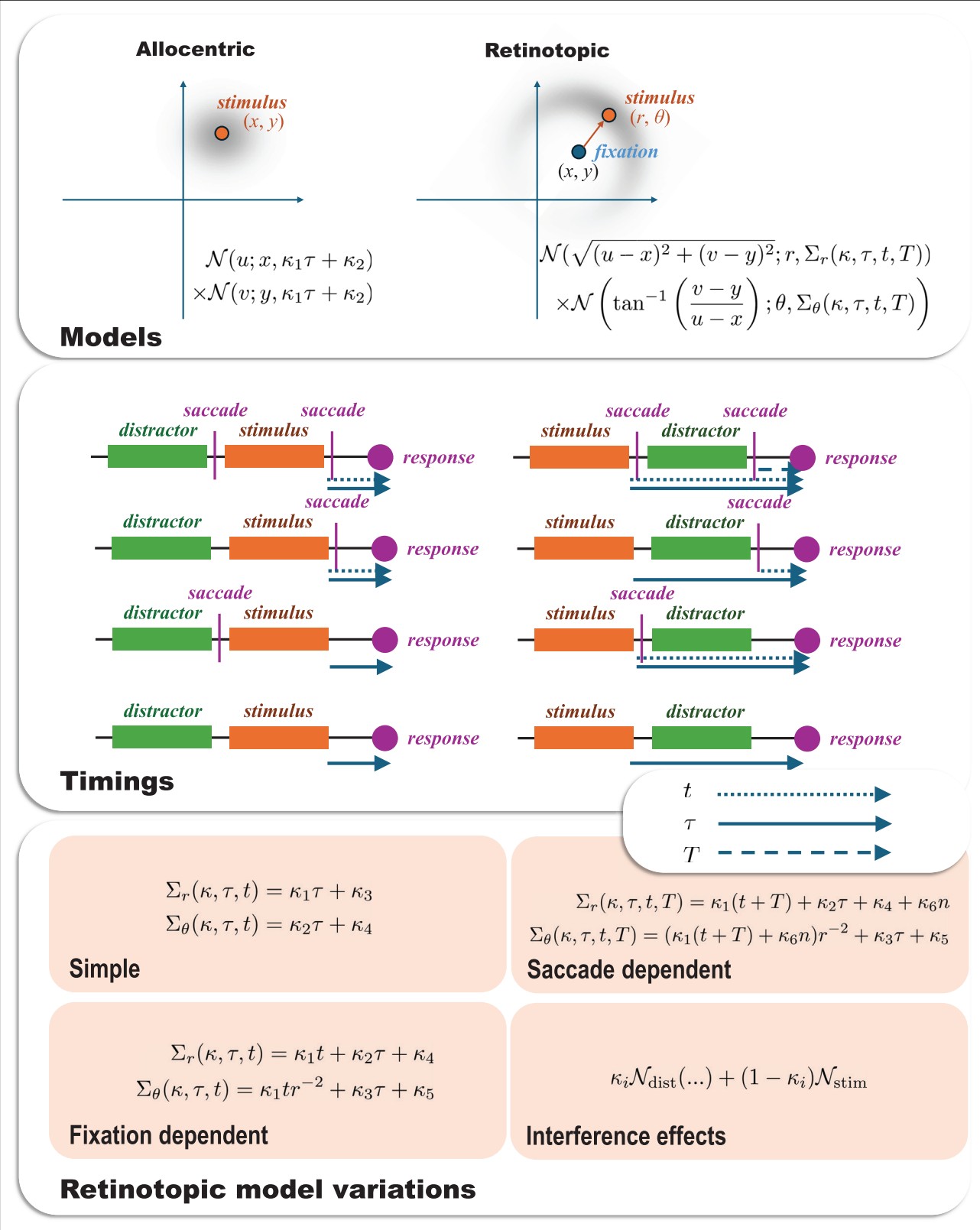

**Figure 9.** Models and hypotheses. This figure illustrates the form of the models used for our model comparisons. The upper panel shows the distinction between allocentric screen-centred coordinates, which we treat as a decay in Euclidean space, such that the probability density of a response (coordinates ($u$,$v$)) is a normal distribution whose covariance grows with time centred on the stimulus. In contrast, the retinotopic representation uses polar coordinates with both radial and angular decays. There are several ways in which the associated covariance can be formulated, and this depends

*Figure 9 continued on next page*

*Figure 9 continued*

upon the different times over which the decay in memory might occur. The different timings in the alternative experimental conditions are shown schematically in the middle panel, dealing with the time since the stimulus was visible ($\tau$), the time since the first saccade (if any) since the stimulus was present ($t$), and the time since the second saccade (if any) after the stimulus ($T$). The variations, based on these timings, for the dynamics of the covariance for the retinotopic model, are shown in the lower panel. The simple model assumes that any decay in the fixation location around which the retinotopic coordinates can be captured by decays in the radial and angular coordinates. This implies there is no difference in the form of the decay when the same fixation location is sustained after the stimulus disappears compared to when a subsequent saccade is made. The fixation-dependent model assumes instead that there is an independent decay in the fixation location, which only begins once we have moved our eyes from the location at which we saw the stimulus. In contrast, the saccade-dependent framing assumes that, rather than retaining a memory of fixation location, we reconstruct the fixation location based upon our memory of the saccade vectors made since that fixation. Here, the n variable represents the number of saccades made since the relevant fixation. The interference effects are mediated by computing the distribution we would expect from a response to the target stimulus and the distribution we would expect from a response to the distractor stimulus (translated into the same retinotopic space) and taking a weighted sum of the two.

We used relatively conservative priors, which were (for all parameters) normal distributions centred on zero, with a spherical covariance of 1/8. This ensured we needed good evidence for the parameters to be estimated as being confidently non-zero. All parameters were log scaling parameters (ensuring the scaling parameters were positive) except for the interference parameter, when present, which was passed through a sigmoid function to constrain the effect of the parameter to lie between zero and one.

Variational Laplace provides approximate Bayesian model evidence values and posterior probability distributions for each parameter, for each model and participant. By accumulating the log model evidence across all participants and applying a softmax function, we arrived at posterior probabilities for each model. We selected the winning model for subsequent steps of the analysis. Our next stage was to use PEB (*Friston et al., 2016*) to assess the effect of diagnostic group on the individual parameter estimates. PEB is effectively a second level linear model. We used the SPM25 implementation spm_dcm_peb.m (*Tierney et al., 2025*).

Finally, to test hypotheses about the mechanisms that explain performance on complex figure drawing, we took the maximum a posteriori estimates (MAP) for all parameters in the winning model for all participants and used these, normalised by their variance, as regressors for a linear model to predict mean-centred copy scores (omitting those scores >2 SD from the mean) and changes in score on immediate recall. Again, we used spm_nlsi_Newton.m to invert this linear model. Following inversion, we computed the predicted scores based upon the MAP values of the regression coefficients and assessed the percentage of empirical variance explained. The percentage of empirical variance explained by the model was assessed using the formula: $100 \times (1 - var(residuals)/var(empirical\_scores))$.

The computational models were fit to the trial-by-trial response data using all 160 trials for each participant. We adopted this inclusive approach because our Bayesian framework explicitly estimates precision (random noise) as a distinct parameter from the saccadic updating cost. This allows the model to partition the total variance, ensuring that even if a specific dataset or clinical cohort exhibits higher overall noise, this is captured by the precision parameter rather than incorrectly inflating the estimates for saccade-driven memory impairment. However, we acknowledge that the reduced number of compliant trials in the PD cohort (averaging approximately 63%) represents a limitation for across-cohort comparisons. Despite this attrition, the absolute number of compliant trials in the PD group (mean approximately 25 per condition) remained sufficient for robust parameter estimation and valid statistical inference.

## Model validation and recovery analysis

The following section provides a detailed technical assessment of the model inversion scheme, focusing on the discriminability of the model space and the identifiability of individual parameters.

Recovery analyses of this sort are typically used prior to collecting data to allow one to determine whether, in principle, the data are useful in disambiguating between hypotheses. In this sense, they have a role analogous to a classical power calculation. However, their utility is limited when used post hoc when data have already been collected, as the question of whether the models can be disambiguated becomes one of whether non-trivial Bayes factors can be identified from those data.

The reason for including a recovery analysis here is not to identify whether the model inversion scheme identifies a 'true' model. The concept of 'true generative models' commits to a strong

philosophical position which is at odds with the 'all models are wrong, but some are useful' perspective held by many in statistics, e.g., *So, 2017*. Of note, one can always confound a model recovery scheme by generating the same data in a simple way, and in (one of an infinite number of) more complex ways. A good model inversion scheme will always recover the simple model and therefore would appear to select the 'wrong' model in a recovery analysis. However, it is still the best explanation for the data. For these reasons, we do not necessarily expect 'good' recoverability in all parameter ranges. This is further confounded by the relationship between the models we have proposed—e.g., an interference model with very low interference will look almost identical to a model with no interference. The important question here is whether they can be disambiguated with real data.

Instead, the value of a post hoc recovery analysis here is to evaluate whether there was a sensible choice of model space—i.e., that it was not a priori guaranteed that a single model (and, specifically, the model we found to be the best explanation for the data) would explain the results of all others. To address this, for each model, we simulated 16 datasets, each of which relied upon parameters sampled from the model priors, which included examples of each of the experimental conditions. We then fit each of these datasets to each of the seven models to construct the confusion matrix shown in the lower panel of *Figure 10*, by accumulating evidence over each of the 16 participants generated according to each 'true' model (columns) for each of the possible explanatory models (rows). This shows that no one model, for the parameter ranges sampled here, explains all other datasets. Interestingly, our 'winning' model in the empirical analysis is not the best explanation for any of the datasets simulated (including its own). This is reassuring, in that it implies this model winning was not a foregone conclusion and is driven by the data—not just the choice of model space.

This problem is an inverse problem—inferring parameters from a non-linear model. We therefore expect a degree of posterior covariance between parameters and, consequently, that they cannot be disambiguated with complete certainty. While some degree of posterior covariance is inherent to inverse models—including established methods like EEG source localisation—the fact that many of the parameters are estimated with posterior densities that do not include their prior expectations implies the data are informative about these.

The advantage of the Bayesian approach we have adopted here is that we can explicitly quantify posterior covariance between these parameters, and therefore the degree to which they can be disambiguated. While the posterior covariance matrices from empirical data are the relevant measure here, we can better understand the behaviour of the model inversion scheme in relation to the specific models used using the model recovery analysis reported in *Figure 10*.

The middle panel of the figure is key, along with the correlation coefficients reported in the figure caption. Here, we see at least a weak positive correlation (in some cases much stronger) for almost all parameters and limited movement from prior expectations for those parameters that are less convincingly recovered. This reinforces that the ability of the scheme to recover parameters is best assessed in terms of the degree of movement of posterior from prior values following fitting to empirical data.

## Statistical analysis

All statistical analyses were conducted in MATLAB R2025a and JASP (*JASP Team, 2024*).

For colour memory, we calculated the absolute angular error, defined as the shortest distance on the colour wheel between the target and the reported colour (range 0 to $\pi$ radians). For the primary statistical analyses, we utilised the mean absolute error per condition for each participant. By analysing these condition-wise means rather than trial-wise raw data, we invoke the central limit theorem, which ensures that the sampling distribution of these means approximates normality. Because the absolute errors in this paradigm were relatively small and did not approach the $\pi$ boundary (*Figure 5B*) even in the clinical cohorts, the data were treated as a continuous measure in our linear ANOVAs and regression models. Moreover, because location and colour recall involve different scales and units, all analyses were performed independently for each feature to avoid cross-dimensional magnitude comparisons.

For correlations, Spearman's rho was reported with two-tailed p-values (alpha = 0.05) and effect size (Fisher's z). Bayesian comparisons were also performed, with Bayes factor 10 ($BF_{10}$) reported when applicable. For ANOVAs, when Mauchly's test of sphericity indicated that the assumption of sphericity was violated (p<0.05), Greenhouse-Geisser correction was used when sphericity was violated. F-stats, p-values, and partial eta-squared were reported. Note that sphericity correction is not available for

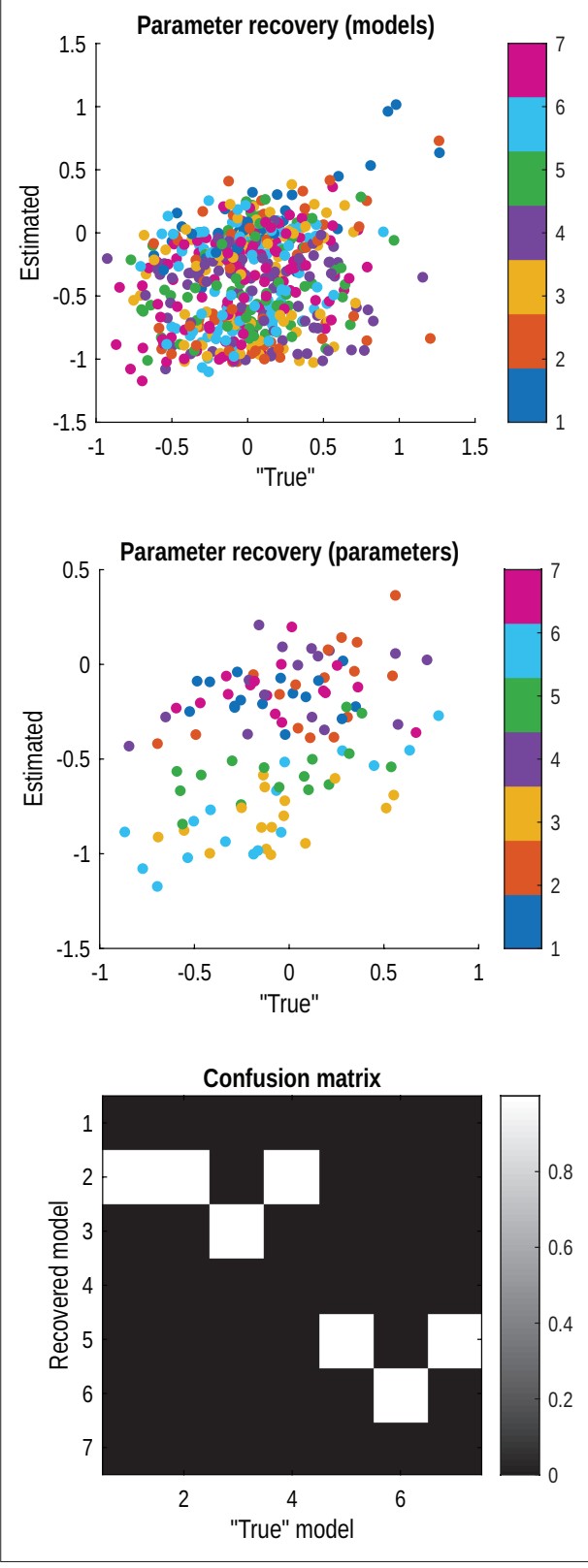

**Figure 10.** Model recovery analysis. This figure reports the results of a model recovery analysis resulting from simulation of data from each model, for 16 participants each (where a participant is generated by sampling parameters from the model priors). Each participant was fit to all models, and the confusion matrix in the lower panel is computed by accumulating the log marginal likelihoods for each combination of model and synthetic

*Figure 10 continued on next page*

*Figure 10 continued*

participant and (softmax) normalising the columns to arrive at posterior probabilities. The upper panel shows all 'true' and estimated parameters recovered when the 'true' models are used on their own datasets. Points are labelled with colours corresponding to the model (see key on the right). Clearly, there is variable veridical recovery across models for the set of sampled parameters. The middle panel presents those estimates for only Model 7 (which in the empirical analysis turns out to be the best explanation). Here, the colours relate to the parameter identities (in the order 'Saccade Decay', 'Radial Decay', 'Angular Decay', 'Radial Encoding', 'Angular Encoding', 'Interference', 'Saccade Encoding'). The correlation coefficients for each of these parameters, respectively, are: −0.62, 0.12, 0.55, 0.35, 0.27, 0.90, 0.17, indicating at least a weak positive correlation for all but the 'Saccade Decay' parameter, and particularly good recovery of the 'Interference' parameter. Of note, those parameters with a weak correlation coefficient do not move far from their (zero) prior expectations, highlighting that those parameters informed by data are those that move from their prior values—something that is straightforward to assess in the empirical analysis. The analysis presented in this figure is somewhat crude and is specific to the set of parameters sampled from the priors, which may or may not reflect real data. However, it gives a useful sense of the behaviour of the model inversion scheme in relation to this specific model (i.e. hypothesis) space.

factors with two levels, such as the item factor (only two items). For multiple comparisons, Bonferroni-corrected p-values (p(bonf)) were reported.

To quantify the evidence for the findings, particularly regarding null effects in the colour domain, we supplemented our frequentist analyses with Bayesian repeated measures ANOVAs. We calculated inclusion Bayes factors ($BF_{incl}$) to determine the evidence for including specific factors across the model space, and $BF_{01}$ to quantify evidence in favour of the null hypothesis. Bayes factors were interpreted using the classification scheme, where BF>3 is considered moderate evidence and BF>100 is considered decisive evidence. All Bayesian analyses were conducted in JASP using default prior distributions.

Multiple linear regression models were fit for each participant using all trials. The regression analyses were conducted via the MATLAB function *regress*. The resulting beta coefficients were then averaged across participants. A one-sample t-test (two-tailed) was used to determine if the group average beta coefficient for each predictor was significantly different from zero. The t-statistic and p-value (two-tailed) are reported for each predictor.

Mediation analysis with multiple mediators was conducted in JASP, using standardised regression coefficients (**JASP Team, 2024**). The bias-corrected bootstrap method with 1000 iterations was used to estimate confidence intervals for the indirect and direct effects (**Biesanz et al., 2010**).

The Bayes factor was computed as the division of accumulated posterior probability between the winning model and the next best alternative.

For comparing group differences in the estimated parameters of the winning model, a PEB approach was utilised, employing spm_dcm_peb in SPM25. This hierarchical Bayesian method allows for robust group-level inference by modelling individual parameter estimates as samples drawn from a group distribution, thereby accounting for inter-subject variability and propagating uncertainty from the first (individual) level. For each parameter of interest, the group-level posterior mean and covariance were obtained from the PEB analysis. To assess statistically significant differences between groups for a given parameter, the probability of a directional difference between the posterior estimates of any two groups was calculated using their respective posterior distributions. A difference was deemed statistically significant if this calculated probability was greater than 0.95 (corresponding to strong evidence in favour of a directional difference).

## Eye-tracking recording

Participants sat in front of a 21" CRT computer screen (1024×768 pixels; 100 Hz refresh rate) at a viewing distance of 60 cm in a dimly lit quiet room with their head supported on a chinrest. Visual stimuli were presented using MATLAB (The MathWorks) and Psychophysics Toolbox (**Brainard, 1997**; **Kleiner et al., 2007**) on a Windows-XP computer. Eye positions during the LOCUS task were monitored using a frame-mounted infrared eye-tracking camera (Eyelink 1000 Tower Mounted, SR Research Ltd.). Right eye was recorded at a sampling rate of 1000 Hz.

Prior to the experiment, a standard nine-point calibration and validation procedure was performed. Participants were instructed to fixate a small black circle with a white centre (0.5°) as it appeared sequentially at nine points forming a 3×3 grid across the screen. Calibration was accepted only if the

mean validation error was below 0.5° and the maximum error at any single point was below 1.0°. If these criteria were not met, or if the experimenter noticed significant gaze drift between blocks, the calibration procedure was repeated. This calibration ensured high spatial accuracy across the entire display area, facilitating the precise monitoring of fixations on item frames and saccadic movements to the response colour wheel.

## Eye-tracking data analysis

All eye-tracking analysis was performed offline. Interval where the eye tracker detected full and partial eye closures was automatically treated as missing data. Furthermore, intervals where pupil diameter was recorded as 0 or changed dramatically immediately pre- or post-blink were also automatically treated as blinks and removed. Gaze positions were epoched relative to stimulus onset (Item 1, Item 2 and colour wheel) and analysed for 1 s post-stimulus onset. Euclidean distance that the gaze travelled to the target was computed and converted to visual degrees. Fixations were defined as gaze remaining within 2.5° of the target for at least 150 ms. Saccades were defined as eye movements exceeding 150° visual angle per second, lasting longer than 10 ms in duration, and traversing a distance greater than 9.2° from the initial fixation. The distance of 9.2° was defined as it is the radius of perifoveal visual field (*Polyak, 1941*; *Sakurai, 2020*).

Task compliance was assessed based on fixation and saccade criteria specific to each condition. For example, in the Saccade-After-Item-1 condition, compliance required fixations on both Item 1 and Item 2, with no saccade exceeding 9.2° after Item 2 measured in the epoch after the onset of colour wheel.

Eye-tracking analysis confirmed high compliance overall, with participants correctly maintaining fixation or executing saccades on the vast majority of trials (83% across all participants). A mixed ANOVA revealed a main effect of group on trial retention ($F(3,80)=8.06$, $p<0.001$, partial $\eta^2=0.23$), primarily due to lower compliance in the PD cohort (YC: 97 ± 4%; EC: 91 ± 10%; AD: 95 ± 5%; PD: 63 ± 38%). Importantly, there was no significant interaction between group and saccade condition ($F(3.36,80)=1.78$, $p=0.15$, partial $\eta^2=0.008$), suggesting that trial attrition was not disproportionately affected by specific task demands in any group.

We acknowledge that this reduced trial count in the PD group represents a limitation for across-cohort comparison. However, the absolute number of compliant trials in the PD group (mean approx. 25 per condition) remained sufficient for robust trial-by-trial parameter estimation. Furthermore, the lack of a significant group-by-condition interaction confirms that the results reported for this cohort remain valid and that our primary finding of a selective spatial memory deficit is robust to these differences in data retention.

## Acknowledgements

This research was supported by funding from the Wellcome Trust and National Institute of Health and Care Research (NIHR) Oxford Health Biomedical Research Centre (BRC). SZ, RU, VK, GDJ, AS, ST, and MH were funded by the Wellcome Trust (206330/Z/17/Z and 226645/Z/22/Z). TP is supported by an NIHR Academic Clinical Fellowship (ref: ACF-2023-13-013). SGM was funded by a Medical Research Council (MRC) Clinician Scientist Fellowship (MR/P00878/X), NIHR BRC and NIHR Oxford Health BRC.

## Additional information

### Funding

| Funder | Grant reference number | Author |
| --- | --- | --- |
| Wellcome Trust | 10.35802/206330 | Sijia Zhao<br>Verena Klar<br>Gabriel Davis Jones<br>Anna Scholcz<br>Sofia Toniolo<br>Masud Husain |

| Funder | Grant reference number | Author |
| --- | --- | --- |
| Wellcome Trust | 10.35802/226645 | Sijia Zhao<br>Sofia Toniolo<br>Masud Husain |
| Medical Research Council | MR/P00878/X | Sanjay G Manohar |
| National Institute for Health and Care Research | ACF-2023-13-013 | Thomas Parr |
| NIHR Oxford Health BRC | | Sofia Toniolo<br>Sanjay G Manohar<br>Masud Husain |
| NIHR Oxford Biomedical Research Centre | | Sanjay G Manohar |

The funders had no role in study design, data collection and interpretation, or the decision to submit the work for publication. For the purpose of Open Access, the authors have applied a CC BY public copyright license to any Author Accepted Manuscript version arising from this submission.

## Author contributions

Sijia Zhao, Software, Formal analysis, Validation, Investigation, Visualization, Methodology, Writing – original draft, Project administration, Writing – review and editing; Thomas Parr, Software, Formal analysis, Validation, Visualization, Methodology, Writing – original draft, Writing – review and editing; Rob Udale, Conceptualization, Resources, Data curation, Software, Investigation, Methodology; Verena Klar, Data curation, Investigation, Project administration; Gabriel Davis Jones, Data curation, Software, Investigation, Project administration; Anna Scholcz, Data curation, Project administration; Sofia Toniolo, Resources, Data curation, Investigation, Project administration, Writing – review and editing; Sanjay G Manohar, Conceptualization, Resources, Supervision, Investigation, Methodology, Writing – review and editing; Masud Husain, Conceptualization, Resources, Supervision, Funding acquisition, Investigation, Project administration, Writing – review and editing

## Author ORCIDs

Sijia Zhao ⃝ https://orcid.org/0000-0002-6246-0702
Thomas Parr ⃝ https://orcid.org/0000-0001-5108-5743
Verena Klar ⃝ https://orcid.org/0000-0003-1901-2485
Sanjay G Manohar ⃝ https://orcid.org/0000-0003-0735-4349

## Ethics

The study was performed in accordance with the ethical standards as laid down in the 1964 Declaration of Helsinki and its later amendments. Ethical approval was granted by the University of Oxford ethics committee (IRAS ID: 248379, Ethics Approval Reference: 18/SC/0448). All participants gave written informed consent prior to the start of the study.

Reviewer #1 (Public review): https://doi.org/10.7554/eLife.109581.3.sa1
Reviewer #2 (Public review): https://doi.org/10.7554/eLife.109581.3.sa2
Reviewer #3 (Public review): https://doi.org/10.7554/eLife.109581.3.sa3
Author response https://doi.org/10.7554/eLife.109581.3.sa4

# Additional files

## Supplementary files
MDAR checklist

## Data availability

All experimental data and healthy participants' Rey-Osterrieth Complex Figure drawings are publicly available on the Open Science Framework (OSF) at https://osf.io/95ecp/. All scripts used for the computational modelling are included in the same repository.

The following dataset was generated:

| Author(s) | Year | Dataset title | Dataset URL | Database and Identifier |
|---|---|---|---|---|
| Zhao S | 2024 | Dynamic updating of spatial working memory across eye movements: a computational investigation of transsaccadic integration | https://osf.io/95ecp/ | Open Science Framework, 95ecp |

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
