## [Editor Report · eLife Assessment]

This study makes an **important** contribution by revealing how saccades selectively disrupt spatial working memory while sparing other object features, and by demonstrating how this mechanism is altered in aging and neurodegeneration. The findings are supported by **convincing** evidence derived from well-controlled eye-tracking experiments and systematic generative model comparisons. Together, the work provides a computationally grounded framework that is of importance for understanding trans-saccadic memory and its clinical relevance.

---

## [Referee Report · Reviewer #1 (Public review)]

Summary:

This study employed a saccade-shifting sequential working memory paradigm, manipulating whether a saccade occurred after each memory array to directly compare retinotopic and transsaccadic working memory for both spatial location and color. Across four participant groups (young and older healthy adults, and patients with Parkinson's disease and Alzheimer's disease), the authors found a consistent saccade-related cost specifically for spatial memory - but not for color - regardless of differences in memory precision. Using computational modeling, they demonstrate that data from healthy participants are best explained by a complex saccade-based updating model that incorporates distractor interference. Applying this model to the patient groups further elucidates the sources of spatial memory deficits in PD and AD. The authors then extend the model to explain copying deficits in these patient groups, providing evidence for the ecological validity of the proposed saccade-updating retinotopic mechanism.

Strengths:

Overall, the manuscript is well written, and the experimental design is both novel and appropriate for addressing the authors' key research questions. I found the study to be particularly comprehensive: it first characterizes saccade-related costs in healthy young adults, then replicates these findings in healthy older adults, demonstrating how this "remapping" cost in spatial working memory is age-independent. After establishing and validating the best-fitting model using data from both healthy groups, the authors apply this model to clinical populations to identify potential mechanisms underlying their spatial memory impairments. The computational modeling results offer a clearer framework for interpreting ambiguities between allocentric and retinotopic spatial representations, providing valuable insight into how the brain represents and updates visual information across saccades. Moreover, the findings from the older adult and patient groups highlight factors that may contribute to spatial working memory deficits in aging and neurological disease, underscoring the broader translational significance of this work.

Comments on revisions:

The authors have addressed my earlier concerns.

---

## [Referee Report · Reviewer #2 (Public review)]

Summary:

Zhao et al investigate how object location and colour are degraded across saccadic eye movements. They employ an eye-tracking task that requires participants to remember two sequentially presented items and subsequently report the colour and position of either one of these. Through counterbalancing of the presence or absence of saccades across items, the authors endeavour to dissect the impact of saccades independently on item location or colour. These behavioural findings form the basis of generative models designed to test competing, nested accounts of how stored information is stored and updated across saccades.

Strengths:

The combination of eye-tracking and generative modelling is a strength of the paper, which opens new perspectives into the impact of Alzheimer's and Parkinson's disease on the performance of visuospatial cognitive tests. The finding that the model parameters covary with clinical performance on the ROCF test is a nice example of a "computational assay" of disease.

Comments on revisions:

I thank the authors for their detailed responses and revisions arising from my feedback on the original manuscript. The revised manuscript adequately addresses all of my concerns.

---

## [Referee Report · Reviewer #3 (Public review)]

Summary:

The manuscript introduces a visual paradigm aimed at studying tran-saccadic memory.

The authors observe how memory of object location is selectively impaired across eye movements, whereas object colour memory is relatively immune to intervening eye movements.

Results are reported for young and elderly healthy controls, as well as PD and AD participants.

A computational model is introduced to account for these results, indicating how early differences in memory encoding and decay (but not tran-saccadic updating per se) can account for the observed differences between healthy controls and clinical groups.

In the revised manuscript, the authors have addressed most of my initial concerns. The dataset is generally compelling, as it includes healthy younger and older adults as well as clinical populations. In addition, the authors propose an interesting modelling approach designed to isolate and characterize the key components underlying the observed patterns of results.

It is important to acknowledge potential limitations of the modelling approach, particularly the differences in the number of parameters across the tested models. As models with more parameters typically achieve better fit, this issue warrants careful consideration. The authors have substantially addressed this point in their rebuttal.

Concerns regarding the specificity of the findings were also raised and have been adequately discussed in the authors' response. Specifically, they clarified the selective impact of saccade-related costs on spatial working memory updating across eye movements-without affecting feature‑based memory (e.g., color) -as well as the specificity of the updating effects observed with the Rey-Osterrieth Complex Figure.

---

## [Author Response]

The following is the authors’ response to the original reviews.

**Reviewer #1 (Public review):**

Thank you so much for your comprehensive and insightful assessment of our manuscript. We appreciate your recognition of the novelty of our experimental design and the utility of our computational framework for interpreting visual remapping across the lifespan and in clinical populations. We are very grateful for your suggestions regarding the narrative flow, which have helped us to improve the manuscript's focus and coherence. Our responses to your specific concerns are detailed below.

(1) Relevance of the figure-copy results (pp. 13-15). Is it necessary to include the figure-copy task results within the main text? The manuscript already presents a clear and coherent narrative without this section. The figure-copy task represents a substantial shift from the LOCUS paradigm to an entirely different task that does not measure the same construct. Moreover, the ROCF findings are not fully consistent with the LOCUS results, which introduces confusion and weakens the manuscript's coherence. While I understand the authors' intention to assess the ecological validity of their model, this section does not effectively strengthen the manuscript and may be better removed or placed in the Supplementary Materials.

We thank the reviewer for their perspective regarding the narrative flow and the transition between the LOCUS paradigm and the ROCF results. However, we remain keen to retain these findings in the main text, as they provide critical ecological and clinical validation for the computational mechanisms identified in our study.

We think these results strengthen the manuscript for the following main reasons:

(1) The ROCF we used is a standard neuropsychological tool for identifying constructional apraxia. Our results bridge the gap between basic cognitive neuroscience and clinical application by demonstrating that specific remapping parameters—rather than general memory precision—predict real-world deficits in patients.

(2) The finding that our winning model explains approximately 62% of the variance in ROCF copy scores across all diagnostic groups further indicates that these parameters from the LOCUS task represent core computational phenotypes that underpin complex, real-life visuospatial construction (copying drawings).

(3) Previous research has often observed only a weak or indirect link between drawing ability and traditional working memory measures, such as digit span (Senese et al., 2020). This was previously attributed to “deictic” strategies—like frequent eye and hand movements—that minimise the need to hold large amounts of information in memory (Ballard et al., 1995; Cohen, 2005; Draschkow et al., 2021). While our study was not exclusively designed to catalogue all cognitive contributions to drawing, the findings provide significant and novel evidence indicating that transsaccadic integration is a critical driver of constructional (copying drawing) ability. By demonstrating this link, the results provide evidence to stimulate a new direction for future research, shifting the focus from general memory capacity toward the precision of spatial updating across eye movements.

In summary, by including the ROCF results in the main text, we provide evidence for a functional role for spatial remapping that extends beyond perceptual stability into the domain of complex visuomotor control. We have expanded on these points throughout the revised manuscript:

In the Introduction: p.2:

“The clinical relevance of these spatial mechanisms is underscored by significant disruptions to visuospatial processing and constructional apraxia—a deficit in copying and drawing figures—observed in neurodegenerative conditions such as Alzheimer's disease (AD) and Parkinson's disease (PD).[20,21] This raises a crucial question: do clinical impairments in complex visuomotor tasks stem from specific failures in transsaccadic remapping? If so, the computational parameters that define normal spatial updating should also provide a mechanistic account of these clinical deficits, differentiating them from general age-related decline.”

p.3: "Finally, by linking these mechanistic parameters to a standard clinical measure of constructional ability (the Rey-Osterrieth Complex Figure task), we demonstrate that transsaccadic updating represents a core computational phenotype underpinning real-world visuospatial construction in both health and neurodegeneration.

In the Results:

“To assess whether the mechanistic parameters derived from the LOCUS task represent core phenotypes of real-world visuospatial abilities, we also instructed all participants to complete the Rey-Osterrieth Complex Figure copy task (ROCF; Figure 7A) on an Android tablet using a digital pen (see examples in Figure 7B; all Copy data are available in the open dataset: https://osf.io/95ecp/). The ROCF is a gold-standard neuropsychological tool for identifying constructional apraxia.[29] Historically, drawing performance has shown only weak or indirect correlations with traditional working memory measures.[30] This disconnect has been attributed to active visual-sampling strategies—frequent eye movements that treat the environment as an external memory buffer, minimising the necessity of holding large volumes of information in internal working memory.[3–5]

We hypothesised that drawing accuracy is primarily constrained by the precision of spatial updating across frequent saccades rather than raw memory capacity. To evaluate the ecological validity of the identified saccade-updating mechanism, we modelled individual ROCF copy scores across all four groups using the estimated (maximum a posteriori) parameters from the winning “Dual (Saccade) + Interference” model (Model 7; Figure 8) as regressors in a Bayesian linear model. Prior to inclusion, each regressor was normalised by dividing by the square root of its variance.

This model successfully explained 61.99% of the variance in ROCF copy scores, indicating that these computational parameters are strong predictors of real-word constructional ability (Figure 8A). … This highlights the critical role of accurate remapping based on saccadic information; even if the core saccadic update mechanism is preserved across groups (as shown in previous analyses), the precision of this updating process is crucial for complex visuospatial tasks. Moreover, worse ROCF copy performance is associated particularly with higher initial angular encoding error. This indicates that imprecision in the initial registration of angular spatial information contributes to difficulties in accurately reproducing complex visual stimuli.”

In the Discussion:

“Importantly, our computational framework establishes a direct mechanistic link between trassaccadic updating and real-world constructional ability. Specifically, higher saccade and angular encoding errors contribute to poorer ROCF copy scores. By mapping these mechanistic estimates onto clinical scores, we found that the parameters derived from our winning model explain approximately 62% of the variance in constructional performance across groups. These findings suggest that the computational parameters identified in the LOCUS task represent core phenotypes of visuospatial ability, providing a mechanistic bridge between basic cognitive theory and clinical presentation.

This relationship provides novel insights into the cognitive processes underlying drawing, specifically highlighting the role of transsaccadic working memoty.ry. Previous research has primarily focused on the roles of fine motor control and eye-hand coordination in this skill.[4,50–55] This is partly because of consistent failure to find a strong relation between traditional memory measures and copying abili [4,31] For instance, common measures of working memory, such as digit span and Corsi block tasks, do not directly predict ROCF copying performance.[31,56] Furthermore, in patients with constructional apraxia, these memory performance measures often remain relatively preserved despite significant drawing impairments.[56–58] In the literature, this lack of association has often been attributed to “deictic” visual-sampling strategies, characterised by frequent eye movements that treat the environment as an external memory buffer, thereby minimising the need to maintain a detailed internal representation.[4,59] In a real-world copying task, the ROCF requires a high volume of saccades, making it uniquely sensitive to the precision of the dynamic remapping signals identified here. Recent eye-tracking evidence confirms that patients with AD exhibit significantly more saccades and longer fixations during figure copying compared to controls, potentially as a compensatory response to trassaccadic working memory constraints.[56] This high-frequency sampling—averaging between 150 and 260 saccades for AD patients compared to approximately 100 for healthy controls—renders the task highly dependent on the precision of dynamic remapping signals.[56] To ensure this relationship was not driven by a general "g-factor" or non-spatial memory impairment, we further investigated the role of broader cognitive performance using the ACE-III Memory subscale. We found that the relationship between transsaccadic working memory and ROCF performance remains highly significant, even after controlling for age, education, and ACE-III Memory subscore. This suggests that transsaccadic updating may represent a discrete computational phenotype required for visuomotor control, rather than a non-specific proxy for global cognitive decline.

In other words, even when visual information is readily available in the world, the act of copying depends critically on working memory across saccades. This reveals a fundamental computational trade-off: while active sampling strategies (characterised with frequent eye-hand movements) effectively reduce the load on capacity-limited working memory, they simultaneously increase the demand for precise spatial updating across eye movements. By treating the external world as an "outside" memory buffer, the brain minimises the volume of information it must hold internally, but it becomes entirely dependent on the reliability with which that information is remapped after each eye movement. This perspective aligns with, rather contradicts, the traditional view of active sampling, which posits that individuals adapt their gaze and memory strategies based on specific task demands.[3,60] Furthermore, this perspective provides a mechanistic framework for understanding constructional apraxia; in these clinical populations, the impairment may not lie in a reduced memory "span," but rather in the cumulative noise introduced by the constant spatial remapping required during the copying process.[58,61]

Beyond constructional ability, these findings suggest that the primary evolutionary utility of high-resolution spatial remapping lies in the service of action rather than perception. While spatial remapping is often invoked to explain perceptual stability,[11–13,15] the necessity of high-resolution transsaccadic memory for basic visual perception is debated.[13,62–64] A prevailing view suggests that detailed internal models are unnecessary for perception, given the continuous availability of visual information in the external world.[13,44] Our findings support an alternative perspective, aligning with the proposal that high-resolution transsaccadic memory primarily serves action rather than perception.[13] This is consistent with the need for precise localisation in eye-hand coordination tasks such as pointing or grasping.[65] Even when unaware of intrasaccadic target displacements, individuals rapidly adjust their reaching movements, suggesting direct access of the motor system to remapping signals.66 Further support comes from evidence that pointing to remembered locations is biased by changes in eye position,[67] and that remapping neurons reside within the dorsal “action” visual pathway, rather than the ventral “perception” visual pathway.[13,68,69] By demonstrating a strong link between transsaccadic working memory and drawing (a complex fine motor skill), our findings suggest that precise visual working memory across eye movements plays an important role in complex fine motor control.”

(2) Model fitting across age groups (p. 9).It is unclear whether it is appropriate to fit healthy young and healthy elderly participants' data to the same model simultaneously. If the goal of the model fitting is to account for behavioral performance across all conditions, combining these groups may be problematic, as the groups differ significantly in overall performance despite showing similar remapping costs. This suggests that model performance might differ meaningfully between age groups. For example, in Figure 4A, participants 22-42 (presumably the elderly group) show the best fit for the Dual (Saccade) model, implying that the Interference component may contribute less to explaining elderly performance.Furthermore, although the most complex model emerges as the best-fitting model, the manuscript should explain how model complexity is penalized or balanced in the model comparison procedure. Additionally, are Fixation Decay and Saccade Update necessarily alternative mechanisms? Could both contribute simultaneously to spatial memory representation? A model that includes both mechanisms-e.g., Dual (Fixation) + Dual (Saccade) + Interference-could be tested to determine whether it outperforms Model 7 to rule out the sole contribution of complexity.

We thank you for the opportunity to expand upon and clarify our modelling approach. Our decision to use a common generative model for both young and older adults was grounded in the empirical finding that there was no significant interaction between age group and saccade condition for either location or colour memory. While older adults demonstrated lower baseline precision, the specific "saccade cost" remained remarkably consistent across cohorts. This was the justification we proceeded on to use of a common model to assess quantitative differences in parameter estimates while maintaining a consistent mechanistic framework for comparison.

Moreover, our winning model nests simpler models as special cases, providing the flexibility to naturally accommodate groups where certain components—such as interference—might play a reduced role. This ultimately confirms that the mechanisms for age-related memory deficits in this task reflect more general decline rather than a qualitative failure of the saccadic remapping process.

This approach is further supported by the properties of the Bayesian model selection (BMS) procedure we used, which inherently penalises the inclusion of unnecessary parameters. Unlike maximum likelihood methods, BMS compares marginal likelihoods, representing the evidence for a model integrated over its entire parameter space. This follows the principle of Bayesian Occam’s Razor, where a model is only favoured if the improvement in fit justifies the additional parameter space; redundant parameters instead "dilute" the probability mass and lower the model evidence.

Consequently, we contend that a hybrid model combining fixation and saccade mechanisms is unnecessary, as we have already adjudicated between alternative mechanisms of equal complexity. Specifically, Model 6 (Dual Fixation + Interference) and Model 7 (Dual Saccade + Interference) possess an identical number of parameters. The fact that Model 7 emerged as the clear winner—providing substantial evidence against Model 6 with a Bayes Factor of 6.11—demonstrates that our model selection is driven by the specific mechanistic account of the data rather than a simple preference for complexity.

We have revised the Results and Discussion sections of the manuscript to state these points more explicitly for readers and have included references to established literature regarding the robustness of marginal likelihoods in guarding against overfitting.

In the Results,

“By fitting these models to the trial-by-trial response data from all healthy participants (N=42), we adjudicated between competing mechanisms to determine which best explained participant performance (Figure 4). We used random-effects Bayesian model selection to identify the most plausible generative model. This process relies on the marginal likelihood (model evidence), which inherently balances model fit against complexity—a principle often referred to as Occam’s razor.[25–27] The analysis yielded a strong result: the “Dual (Saccade) + Interference” model (Model 7 in Table 1) emerged as the winning model, providing substantial evidence against the next best alternative with a Bayes Factor of 6.11.”

In the Discussion:

“Our framework employs Variational Laplace, a method used to recover computational phenotypes in clinical populations like those with substance use disorders,[34,35] and the models we fit using this procedure feature time-dependent parameterisation of variance—conceptually similar to the widely-used Hierarchical Gaussian Filter.[36–39] Importantly, the risk of overfitting is mitigated by the Bayesian Model Selection framework; by utilising the marginal likelihood for model comparison, the procedure inherently penalises excessive model complexity and promotes generalisability.[25–27,40] This generalisability was further evidenced by the model's ability to predict performance on the independent ROCF task, confirming that these parameters represent robust mechanistic phenotypes rather than idiosyncratic fits to the initial dataset.”

Minor point: On p. 9, line 336, Figure 4A does not appear to include the red dashed vertical line that is mentioned as separating the age groups.

Thank you for pointing out this inconsistency. We apologise for the oversight; upon further review, we concluded that the red dashed vertical line was unnecessary for the clear presentation of the data. We have therefore removed the line from Figure 4A and deleted the corresponding sentence in the figure caption.

(3) Clarification of conceptual terminology.Some conceptual distinctions are unclear. For example, the relationship between "retinal memory" and "transsaccadic memory," as well as between "allocentric map" and "retinotopic representation," is not fully explained. Are these constructs related or distinct? Additionally, the manuscript uses terms such as "allocentric map," "retinotopic representation," and "reference frame" interchangeably, which creates ambiguity. It would be helpful for the authors to clarify the relationships among these terms and apply them consistently.

Thank you for pointing this out. We have revised the manuscript to ensure that these terms are applied with greater precision and consistency. Our revisions standardise the terminology based on the following distinctions:

Reference frames: We distinguish between the eye-centred reference frame (coordinate systems that shift with gaze) and the world-centred reference frame (coordinate systems anchored to the environment).

Retinotopic representation vs. allocentric map: We clarify that retinotopic representations are encoded within an eye-centred reference frame and are updated with every ocular movement. Conversely, the allocentric map is anchored to stable environmental features, remaining invariant to the observer’s gaze direction or position.

Retinotopic memory vs. transsaccadic memory: We have removed the term "retinal memory" to avoid ambiguity. We now consistently use retinotopic memory to describe the persistence of visual information in eye-centred coordinates within a single fixation. In contrast, transsaccadic memory refers to the higher-level integration of visual information across saccades, which involves the active updating or remapping of representations to maintain stability.

To incorporate these clarifications, we have implemented the following changes:

In the Introduction, the second paragraph has been entirely rewritten to establish these definitions at the outset, providing a clearer theoretical framework for the study.

“Central to this enquiry is the nature of the coordinate system used for the brain's internal spatial representation. Does the brain maintain a single, world-centred (allocentric) map, or does it rely on a dynamic, eye-centred (retinotopic) representation?[11,13,15,16] In the latter system, retinotopic memory preserves spatial information within a fixation, whereas transsaccadic memory describes the active process of updating these representations across eye movements to achieve spatiotopic stability—the perception of a stable world despite eye movements.[11,16–18] If spatial stability is indeed reconstructed through such remapping, the mechanism remains unresolved: do we retain memories of absolute fixation locations, or do we reconstruct these positions from noisy memories of the intervening saccade vectors? We can test these hypotheses by analysing when and where memory errors occur. Assuming that memory precision declines over time,[19] the resulting error distributions should reveal the specific variables that are represented and updated across each saccade.”

In the Results, the opening section of the Results has been reorganised to align with this terminology. We have ensured that the hypotheses and behavioural data—specifically the definition of "saccade cost"—are introduced using this consistent conceptual vocabulary to improve the overall coherence of the narrative.

(4) Rationale for the selective disruption hypothesis (p. 4, lines 153-154). The authors hypothesize that "saccades would selectively disrupt location memory while leaving colour memory intact." Providing theoretical or empirical justification for this prediction would strengthen the argument.

We have revised the Results to state the hypothesis more explicitly and expanded the Discussion to provide a robust theoretical and empirical rationale:

In the Results,

“This design allowed us to isolate and quantify the unique impact of saccades on spatial memory, enabling us to test competing hypotheses regarding spatial representation. If spatial memory were solely underpinned by an allocentric mechanism, precision should remain comparable across all conditions as the representation would be world-centred and unaffected by eye movements. Thus, performance in the no-saccade condition should be comparable to the two-saccade condition. Conversely, if spatial memory relies on a retinotopic representation requiring active updating across eye movements, the two-saccade condition was anticipated to be the most challenging due to cumulative decay in the memory traces used for stimulus reconstruction after each saccade.[22] Critically, we hypothesised that this saccade cost would be specific to the spatial domain; while location requires active remapping via noisy oculomotor signals, non-spatial features like colour are not inherently tied to coordinate transformations and should therefore remain stable (see more in Discussion below).

Meanwhile, the no-saccade condition was expected to yield the most accurate localisation, relying solely on retinotopic information (retinotopic working memory). These predictions were confirmed in young healthy adults (N = 21, mean age = 24.1 years, ranged between 19 and 34). A repeated measures ANOVA revealed a significant main effect of saccades on location memory (F(2.2,43.9)=33.2, p<0.001, partial η²=0.62), indicating substantial impairment after eye movements (Figure 2A). In contrast, colour memory remained remarkably stable across all saccade conditions (Figure 2B; F(2.2, 44.7) = 0.68, p=0.53, partial η² = 0.03).

This “saccade cost”—the loss of memory precision following an eye movement—indicates that spatial representations require active updating across saccades rather than being maintained in a static, world-centred reference frame.

Critically, our comparison between spatial and colour memory does not rely on the absolute magnitude of errors, which are measured in different units (degrees of visual angle vs. radians). Instead, we assessed the relative impact of the same saccadic demand on each feature within the same trial. While location recall showed a robust saccade cost, colour recall remained statistically unchanged. To ensure this null effect was not due to a lack of measurement sensitivity, we examined the recency effect; recall performance for the second item was predicted to be better than for the first stimulus in each condition.[23,24] As expected, colour memory for Item 2 was significantly more accurate than for Item 1 (F(1,20) = 6.52, p = 0.02, partial η² = 0.25), demonstrating that the task was sufficiently sensitive to detect standard working memory fluctuations despite the absence of a saccade-induced deficit.”

In the Discussion, we now write that on p.18:

“A clear finding was the specificity of the saccade cost to spatial features; it was not observed for non-spatial features like colour, even in neurodegenerative conditions. This discrepancy challenges notions of fixed visual working memory capacity unaffected by saccades.16,44–46 The differential impact on spatial versus non-spatial features in transsaccadic memory aligns with the established "what" and "where" pathways in visual processing.32,33 For objects to remain unified, object features must be bound to stable representations of location across saccades.19 One possibility is that remapping updates both features and location through a shared mechanism, predicting equal saccadic interference for both colour and location in the present study.

However, our findings suggest otherwise. One potential concern is whether this dissociation simply reflects the inherent spatial noise introduced by fixational eye movements (FEMs), such as microssacades and drifts.47 Because locations are stored in a retinotopic frame, fixational instability necessarily shifts retinal coordinates over time. However, the "saccade cost" here was defined as the error increase relative to a no-saccade baseline of equal duration; because both conditions are subject to the same fixational drift, any FEM-induced noise is effectively subtracted out. Thus, despite the ballistic and non-Gaussian nature of FEMs,48 they cannot account for the fact the saccade cost in the spatial memory, but total absence in the colour domain. Another possibility is that this dissociation reflects differences in baseline task difficulty or dynamic range. Yet, the presence of a robust recency effect in colour memory (Figure 2B) confirms that our paradigm was sensitive to memory-dependent variance and was not limited by floor or ceiling effects.

The fact that identical eye movements—executed simultaneously and with identical vectors—systematically degraded spatial precision while sparing colour suggests a feature-specific susceptibility to transsaccadic remapping. This supports the view that the computational process of updating an object’s location involves a vector-subtraction mechanism—incorporating noisy oculomotor commands (efference copies)—that introduces specific spatial variance. Because this remapping is a coordinate transformation, the resulting sensorimotor noise does not functionally propagate to non-spatial feature representations. Consequently, features like colour may be preserved or automatically remapped without the precision loss associated with spatial updating.11,49 Our paradigm thus provides a refined tool to investigate the architecture of transsaccadic working memory across distinct object features.”

(5) Relationship between saccade cost and individual memory performance (p. 4, last paragraph).The authors report that larger saccades were associated with greater spatial memory disruption. It would be informative to examine whether individual differences in the magnitude of saccade cost correlate with participants' overall/baseline memory performance (e.g. their memory precision in the no-saccade condition). Such analyses might offer insights into how memory capacity/ability relates to resilience against saccade-induced updating.

We have now conducted the correlation analysis to determine whether baseline memory capacity (no-saccade condition) predicts resilience to saccade-induced updating. The results indicate that these two factors are independent.

To clarify the nature of the saccade-induced impairment, we have updated the text as follows:

p.4: “This “saccade cost”—the loss of memory precision following an eye movement—indicates that spatial representations require active updating across saccades rather than being maintained in a static, world-centred reference frame.”

p.5: “Further analysis examined whether individual differences in baseline memory precision (no-saccade condition) predicted resilience to saccadic disruption. Crucially, individual saccade costs (defined as the precision loss relative to baseline) did not correlate with baseline precision (rho = 0.20, p = 0.20). This suggests that the noise introduced by transsaccadic remapping acts as an independent, additive source of variance that is not modulated by an individual’s underlying memory capacity. These findings imply a functional dissociation between the mechanisms responsible for maintaining a representation and those involved in its coordinate transformation.”

(6) Model fitting for the healthy elderly group to reveal memory-deficit factors (pp. 11-12). The manuscript discusses model-based insights into components that contribute to spatial memory deficits in AD and PD, but does not discuss components that contribute to spatial memory deficits in the healthy elderly group. Given that the EC group also shows impairments in certain parameters, explaining and discussing these outcomes of the EC group could provide additional insights into age-related memory decline, which would strengthen the study's broader conclusions.

This is a very good point. We rewrote the corresponding results section (p.12-13):

“Modelling reveals the sources of spatial memory deficits in healthy aging and neurodegeneration - To understand the source of the observed deficits, we applied the winning ‘Dual (Saccade) + Interference’ model the data from all participants (YC, EC, AD, and PD). By fitting the model to the entire dataset, we obtained estimates of the parameters for each individual, which then formed the basis for our group-level analysis. To formally test for group differences, we used Parametric Empirical Bayes (PEB), a hierarchical Bayesian approach that compares parameter estimates across groups while accounting for the uncertainty of each estimate [28]. This allowed us to identify which specific cognitive mechanisms, as formalised by the model parameters, were affected by age and disease.

The Bayesian inversion used here allows us to quantify the posterior mode and variance for each parameter and the covariance for each parameter. From these, we can compute the probabilities that pairs of parameters differ from one another, which we report as P(A>B)—meaning the posterior probability that the parameter for group A was greater than that for group B.

We first examined the specific parameters differentiating healthy elderly (EC) from young controls (YC) to isolate the factors contributing to non-pathological, age-related decline. The analysis revealed that healthy ageing is primarily characterised by a significant increase in Radial Decay (P(EC > YC) = 0.995), a heightened susceptibility to Interference (P(EC > YC) = 1.000), and a reduction in initial Angular Encoding precision (P(YC < EC) = 0.002; Figure 6). These results suggest that normal ageing degrades the fidelity of the initial memory trace and its resilience over time, while the core computational process of updating information across saccades remains intact.

Beyond these baseline ageing effects, our clinical cohorts exhibited more severe and condition-dependent impairments. Radial decay showed a clear, graded impairment: AD patients had a greater decay rate than PD patients (P(AD > PD) = 1.000), who in turn were more impaired than the EC group (P(PD > EC) = 0.996). A similar graded pattern was observed for Interference, where AD patients were most susceptible (P(AD > PD) = 0.999), while the PD and EC groups did not significantly differ (P(PD > EC) = 0.532).

Patients with AD also showed a tendency towards greater angular decay than controls (P(AD > EC) = 0.772), although this fell below the 95% probability threshold. This effect was influenced by a lower decay rate in the PD group compared to the EC group (P(PD < EC) = 0.037). In contrast, group differences in encoding were less pronounced. While YC exhibited significantly higher precision than all other groups, AD patients showed significantly higher angular encoding error than PD patients (P(AD > PD) = 0.985), though neither group differed significantly from the EC group.

Crucially, parameters related to the saccade itself—saccade encoding and saccade decay—did not differentiate the groups. This indicates that neither healthy ageing nor the early stages of AD and PD significantly impair the fundamental machinery for transsaccadic remapping. Instead, the visuospatial deficits in these conditions arise from specific mechanistic failures: a faster decay of radial position information and increased susceptibility to interference, both of which are present in healthy ageing but significantly amplified by neurodegeneration.”

In the Discussion, we added:

“Although saccade updating was an essential component of the winning model, its two key parameters—initial encoding error and decay rate during maintenance—did not significantly differ across groups. This indicates that the core computational process of updating spatial information based on eye movements is largely preserved in healthy aging and neurodegeneration.

Instead, group differences were driven by deficits in angular encoding error (precision of initial angle from fixation), angular decay, radial decay (decay in memory of distance from fixation), and interference susceptibility. This implies a functional and neuroanatomical dissociation: while the ventral stream (the “what” pathway) shows an age-related decline in the quality and stability of stored representations, the dorsal-stream (the “where” pathway) parietal-frontal circuits responsible for coordinate transformations remain functionally robust.[31–34] These spatial updating mechanisms appear resilient to the normal ageing trajectory and only break down when challenged by the specific pathological processes seen in Alzheimer’s or Parkinson’s disease.”

(7) Presentation of saccade conditions in Figure 5 (p. 11). In Figure 5, it may be clearer to group the four saccade conditions together within each patient group. Since the main point is that saccadic interference on spatial memory remains robust across patient groups, grouping conditions by patient type rather than intermixing conditions would emphasize this interpretation.

There are several valid ways to present these plots, but we chose this format because it allows for a direct visual comparison of the post-hoc group differences within each specific task demand. This arrangement clearly illustrates the graded impairment from young controls through to patients with Alzheimer’s disease across every condition. This structure also directly mirrors our two-way ANOVA, which identified significant main effects for both Group and Condition, but crucially, no significant Group x Condition interaction. We felt that grouping the data by participant group would force readers to look across four separate clusters to compare the slopes, making the stability of the saccadic remapping mechanism much harder to grasp at a glance.

**Reviewer #1 (Recommendations for the authors):**
(1) Formatting of statistical parameters.The formatting of statistical symbols should be consistent throughout the manuscript. Some instances of F, p, and t are italicized, while others are not. All statistical symbols should be italicized.

Thank you for pointing this out. We have audited the manuscript. While we have revised the text to address these instances throughout the Results and Methods sections, any remaining minor formatting inconsistencies will be corrected during the final typesetting stage.

(2) Minor typographical issues.(a) Line 532: "are" should be "be."(b) Line 654: "cantered" should be "centered."(c) Line 213: In "(p(bonf) < 0.001, |t| {greater than or equal to} 5.94)," the t value should be reported with its degrees of freedom, and t should be reported before p. The same applies to line 215.

Thank you for your careful reading. All corrected.

**Reviewer #2 (Public review):**

We thank you for your positive feedback regarding our eye-tracking methodology and computational approach. We appreciate your critical insights into the feature-specific disruption hypothesis and the task structure. We have substantially revised the results and discussion about the saccadic interference on colour memory. Below we will answer your suggestions point-by-point:

**Reviewer #2 (Recommendations for the authors):**
(1) The study treats colour and location errors as comparable when arguing that saccades selectively disrupt spatial but not colour memory. However, these measures are defined in entirely different units (degrees of visual angle vs radians on a colour wheel) and are not psychophysically or statistically calibrated. Baseline task difficulty, noise level, or dynamic range do not appear to be calibrated or matched across features. As a result, the null effect of saccades on colour could reflect lower sensitivity or ceiling effects rather than implicit feature-specific robustness.

We agree that direct comparisons of absolute error magnitudes across different dimensions are not appropriate. Our argument for feature-specific disruption relies not on the scale of errors, but on the presence or absence of a saccade cost within identical trials. In our within-subject design, the same saccade vectors produced a systematic increase in location error while leaving colour error statistically unchanged. To address sensitivity, we observed that colour memory was sufficiently precise to show a significant recency effect (p = 0.02). To further quantify the evidence for the null effect, we performed Bayesian repeated measures ANOVAs, which yielded a BF10 = 0.22. This provides substantial evidence that saccades do not disrupt colour precision, regardless of baseline sensitivity.

We have substantially revised this in Results, Methods and Discussion:

In the Results:

“This design allowed us to isolate and quantify the unique impact of saccades on spatial memory, enabling us to test competing hypotheses regarding spatial representation. If spatial memory were solely underpinned by an allocentric mechanism, precision should remain comparable across all conditions as the representation would be world-centred and unaffected by eye movements. Thus, performance in the no-saccade condition should be comparable to the two-saccade condition. Conversely, if spatial memory relies on a retinotopic representation requiring active updating across eye movements, the two-saccade condition was anticipated to be the most challenging due to cumulative decay in the memory traces used for stimulus reconstruction after each saccade.[22] Critically, we hypothesised that this saccade cost would be specific to the spatial domain; while location requires active remapping via noisy oculomotor signals, non-spatial features like colour are not inherently tied to coordinate transformations and should therefore remain stable (see more in Discussion below).

Meanwhile, the no-saccade condition was expected to yield the most accurate localisation, relying solely on retinotopic information (retinotopic working memory). These predictions were confirmed in young healthy adults (N = 21, mean age = 24.1 years, ranged between 19 and 34). A repeated measures ANOVA revealed a significant main effect of saccades on location memory (F(2.2,43.9)=33.2, p<0.001, partial η²=0.62), indicating substantial impairment after eye movements (Figure 2A). In contrast, colour memory remained remarkably stable across all saccade conditions (Figure 2B; F(2.2, 44.7) = 0.68, p=0.53, partial η² = 0.03).

This “saccade cost”—the loss of memory precision following an eye movement—indicates that spatial representations require active updating across saccades rather than being maintained in a static, world-centred reference frame.

Critically, our comparison between spatial and colour memory does not rely on the absolute magnitude of errors, which are measured in different units (degrees of visual angle vs. radians). Instead, we assessed the relative impact of the same saccadic demand on each feature within the same trial. While location recall showed a robust saccade cost, colour recall remained statistically unchanged. To ensure this null effect was not due to a lack of measurement sensitivity, we examined the recency effect; recall performance for the second item was predicted to be better than for the first stimulus in each condition.[23,24] As expected, colour memory for Item 2 was significantly more accurate than for Item 1 (F(1,20) = 6.52, p = 0.02, partial η² = 0.25), demonstrating that the task was sufficiently sensitive to detect standard working memory fluctuations despite the absence of a saccade-induced deficit.”

In the Methods, at the beginning of “Statistical Analysis”, we added

“Because location and colour recall involve different scales and units, all analyses were performed independently for each feature to avoid cross-dimensional magnitude comparisons.” (p25)

In the Discussion, we added:

“A potential concern is whether the observed dissociation between colour and location reflects differences in baseline task difficulty or dynamic range. Yet, the presence of a robust recency effect in colour memory (Figure 2B) confirms that our paradigm was sensitive to memory-dependent variance and was not limited by floor or ceiling effects.”

(2) Colour and then location are probed serially, without a counter-balanced order. This fixed response order could introduce a systematic bias because location recall is consistently subject to longer memory retention intervals and cognitive interference from the colour decision. The observed dissociation-saccades impair location but not colour, and may therefore reflect task structure rather than implicit feature-specific differences in trans-saccadic memory.

Thank you for the insightful observation regarding our fixed response order. We acknowledge that that a counterbalanced design is typically preferred to mitigate potential order effects. However, we chose this consistent sequence to ensure the task remained accessible for cognitively impaired patients (i.e., the Alzheimer’s disease (AD) and Parkinson’s disease (PD) cohorts). Conducting an eye-tracking memory task with cognitively impaired patients is challenging, as they may struggle with task engagement or forget complex instructions. During the design phase, we prioritised a consistent structure to reduce the cognitive load and task-switching demands that typically challenge these cohorts.

Critically, because the saccade cost is a relative measure calculated by comparing conditions with identical timings, any bias from the fixed order is present in both the baseline and saccade trials. The disruption we report is therefore a specific effect of eye movements that goes beyond the noise introduced by the retention interval or the preceding colour report.

We added the following text in the Methods – experimental procedure (p.22):

“Recall was performed in a fixed order, with colour reported before location. This sequence was primarily chosen to minimise cognitive load and task-switching demands for the two neurological patient cohorts, ensuring the paradigm remained accessible for individuals with AD and PD. While this order results in a slightly longer retention interval for location recall, the saccade cost was identified by comparing location error across experimental conditions with similar timings but varying saccadic demands.”

(3) Relatedly, because spatial representations are retinotopic, fixational eye movements (FEMs - microsaccades and drift) displace the retinal coordinates of encoded positions, increasing apparent spatial noise with time delays. Colour memory, however, is feature-based and unaffected by small retinal translations. Thus, any between-condition or between-group differences in FEMs could selectively inflate location error and the associated model parameters (encoding noise, decay, interference), while leaving colour error unchanged. Note that FEMs tend to be slightly ballistic [1,2], hence not well modelled with a Gaussian blur.

This is a very insightful point. We have now addressed this in detail within the discussion:

“However, our findings suggest otherwise. One potential concern is whether this dissociation simply reflects the inherent spatial noise introduced by fixational eye movements (FEMs), such as microssacades and drifts.[46] Because locations are stored in a retinotopic frame, fixational instability necessarily shifts retinal coordinates over time. However, the "saccade cost" here was defined as the error increase relative to a no-saccade baseline of equal duration; because both conditions are subject to the same fixational drift, any FEM-induced noise is effectively subtracted out. Thus, despite the ballistic and non-Gaussian nature of FEMs,n [47] they cannot account for the fact the saccade cost in the spatial memory, but total absence in the colour domain. Another possibility is that this dissociation reflects differences in baseline task difficulty or dynamic range. Yet, the presence of a robust recency effect in colour memory (Figure 2B) confirms that our paradigm was sensitive to memory-dependent variance and was not limited by floor or ceiling effects.”

(4) There is no in silico demonstration that the modelling framework can recover the true generating model from synthetic data or recover accurate parameters under realistic noise levels, which can be challenging in generative models with a hierarchical structure (as per [3], for example). Figure 8b shows that the parameters possess substantial posterior covariance, which raises concerns as to whether they can be reliably disambiguate.

Many thanks for this comment. We have added a simple recovery analysis as detailed below but are also keen to ensure we fully answer your question—which has more to do with empirical rather than simulated data—and make clear the rationale for this analysis in this instance.

We added this in Supplementary Materials:

“Model validation and recovery analysis

The following section provides a detailed technical assessment of the model inversion scheme, focusing on the discriminability of the model space and the identifiability of individual parameters.

Recovery analyses of this sort are typically used prior to collecting data to allow one to determine whether, in principle, the data are useful in disambiguating between hypotheses. In this sense, they have a role analogous to a classical power calculation. However, their utility is limited when used post-hoc when data have already been collected, as the question of whether the models can be disambiguated becomes one of whether non-trivial Bayes factors can be identified from those data.

The reason for including a recovery analysis here is not to identify whether the model inversion scheme identifies a ‘true’ model. The concept of ‘true generative models’ commits to a strong philosophical position which is at odds with the ‘all models are wrong, but some are useful’ perspective held by many in statistics, e.g., (So, 2017). Of note, one can always confound a model recovery scheme by generating the same data in a simple way, and in (one of an infinite number of) more complex ways. A good model inversion scheme will always recover the simple model and therefore would appear to select the ‘wrong’ model in a recovery analysis. However, it is still the best explanation for the data. For these reasons, we do not necessarily expect ‘good’ recoverability in all parameter ranges. This is further confounded by the relationship between the models we have proposed—e.g., an interference model with very low interference will look almost identical to a model with no interference. The important question here is whether they can be disambiguated with real data.

Instead, the value of a post-hoc recovery analysis here is to evaluate whether there was a sensible choice of model space—i.e., that it was not a priori guaranteed that a single model (and, specifically, the model we found to be the best explanation for the data) would explain the results of all others. To address this, for each model, we simulated 16 datasets, each of which relied upon parameters sampled from the model priors, which included examples of each of the experimental conditions. We then fit each of these datasets to each of the 7 models to construct the confusion matrix shown in the lower panel of Supplementary Figure 3, by accumulating evidence over each of the 16 participants generated according to each ‘true’ model (columns) for each of the possible explanatory models (rows). This shows that no one model, for the parameter ranges sampled here, explains all other datasets. Interestingly, our ‘winning’ model in the empirical analysis is not the best explanation for any of the datasets simulated (including its own). This is reassuring, in that it implies this model winning was not a foregone conclusion and is driven by the data—not just the choice of model space.”

Your point about the posterior covariance is well founded. As we describe in Supplementary Materials, this is an inherent feature of inverse problems (analogous to EEG source localisation). However, the fact that our posterior densities move significantly away from the prior expectations demonstrates that the data are indeed informative. By adopting a Bayesian framework, we are able to explicitly quantify this uncertainty rather than ignoring it, providing a more transparent account of parameter identifiability. We have added the following in the same section of Supplementary Materials:

“This problem is an inverse problem—inferring parameters from a non-linear model. We therefore expect a degree of posterior covariance between parameters and, consequently, that they cannot be disambiguated with complete certainty. While some degree of posterior covariance is inherent to inverse models—including established methods like EEG source localisation—the fact that many of the parameters are estimated with posterior densities that do not include their prior expectations implies the data are informative about these.

The advantage of the Bayesian approach we have adopted here is that we can explicitly quantify posterior covariance between these parameters, and therefore the degree to which they can be disambiguated. While the posterior covariance matrices from empirical data are the relevant measure here, we can better understand the behaviour of the model inversion scheme in relation to the specific models used using the model recovery analysis reported in Supplementary figure 3.

The middle panel of the figure is key, along with the correlation coefficients reported in the figure caption. Here, we see at least a weak positive correlation (in some cases much stronger) for almost all parameters and limited movement from prior expectations for those parameters that are less convincingly recovered. This reinforces that the ability of the scheme to recover parameters is best assessed in terms of the degree of movement of posterior from prior values following fitting to empirical data.”

(5) The authors employ Bayes factors (BFs) to disambiguate models, but BFs would also strengthen the claims that location, but not colour, is impacted by saccades. Despite colour being a circular variable, colour error is analysed using ANOVA on linearised differences (radians). The authors should also arguably use circular statistics, such as the von Mises distribution, for the analysis of colour.

Regarding the use of circular statistics, you are correct that such error distributions are not suitable for ANOVA, and it is better to use circular statistics. However, for the present dataset, we used the mean absolute angular error per condition (ranging from 0 to π radians), which represents the shortest distance on the colour wheel between the target and the response.

This approach effectively linearises the measure by removing the 2π wrap-around boundary. because the observed errors were relatively small and did not cluster near the π boundary—even in the patient cohorts (Figure 5B)—the "wrap-around" effect of circular space is negligible. Moreover, by analysing the mean error across trials for each condition, rather than trial-wise data, we invoke the Central Limit Theorem. This ensures that the distribution of these means is approximately normal, satisfying the fundamental assumptions of ANOVA. Due to these reasons, we adopted simpler linear models. We confirmed that the data did not violate the assumptions of linear statistics. In this low-noise regime, linear and circular models converge on the same conclusions. This has been revised in Methods:

“For colour memory, we calculated the absolute angular error, defined as the shortest distance on the colour wheel between the target and the reported colour (range 0 to π radians). For the primary statistical analyses, we utilised the mean absolute error per condition for each participant. By analysing these condition-wise means rather than trial-wise raw data, we invoke the Central Limit Theorem, which ensures that the sampling distribution of these means approximates normality. Because the absolute errors in this paradigm were relatively small and did not approach the π boundary (Figure 5B) even in the clinical cohorts, the data were treated as a continuous measure in our linear ANOVAs and regression models. Moreover, because location and colour recall involve different scales and units, all analyses were performed independently for each feature to avoid cross-dimensional magnitude comparisons.”

We have also now integrated Bayesian repeated measures ANOVA throughout the manuscript. The Results section for the young healthy adults now reads (p. 4):

“A repeated measures ANOVA revealed a significant main effect of saccades on location memory (F(3, 20) = 51.52, p < 0.001, partial η²=0.72), with Bayesian analysis providing decisive evidence for the inclusion of the saccade factor (BF_incl_ = 3.52 x 10^13, P(incl|data) = 1.00). In contrast, colour memory remained remarkably stable across all saccade conditions (F(3, 20) = 0.57, p = 0.64, partial η² = 0.03). This null effect was supported by Bayesian analysis, which provided moderate evidence in favour of the null hypothesis (BF_01_ = 8.46, P(excl|data) = 0.89), indicating that the data were more than eight times more likely under the null model than a model including saccade-related impairment.”

For elderly healthy adults:

“In contrast, colour memory remained unaffected by saccade demands (F(3, 20) = 0.57, p = 0.65, partial η² = 0.03), again supported by the Bayesian analysis: BF_01_ = 8.68, P(excl|data) = 0.90.”

For patient cohorts:

“Bayesian repeated measures ANOVAs further supported this dissociation, providing moderate evidence for the null hypothesis in the AD group (BF_01_ = 3.35, P(excl|data) = 0.77) and weak evidence in the PD group (BF_01_ = 2.23, P(excl|data) = 0.69). This indicates that even in populations with established neurodegeneration, the detrimental impact of eye movements is specific to the spatial domain.”

Related description is also updated in Methods – Statistical Analysis.

Minor:(1) The modelling is described as computational but is arguably better characterised as a heuristic generative model at Marr's algorithmic level. It does not derive from normative computational principles or describe an implementation in neural circuits.

We appreciate your perspective on the classification of our model within Marr’s hierarchy. We agree that our framework is best characterised as an algorithmic-level generative model. Our objective was to identify the mechanistic principles governing transsaccadic updating rather than to provide a normative derivation or a specific circuit-level implementation.

To ensure readers do not over-interpret the term ‘computational’, we have added a clarifying statement in the Discussion acknowledging the algorithmic nature of the model. Interestingly, we note that a model predicated on this form of spatial diffusion implies a neural field representation with a spatial connectivity kernel whose limit approximates the second derivative of a Dirac delta function. While a formal neural field implementation is beyond the scope of the present work, our algorithmic results provide the necessary constraints for such future biophysical models.

p.20: “While we describe the present framework as 'computational', it is more precisely characterised as an algorithmic-level generative model within Marr’s hierarchy. Our focus was on defining the rules of spatial integration and the sources of eye-movement-induced noise, rather than deriving these processes from normative principles or defining their specific neural implementation.”

(2) I did not find a description of the recruitment and characterization of the AD and PD patients.

Apologies for this omission. We have now included a detailed description of participant recruitment and clinical characterisation in the Methods section and also updated Table 2:

“A total of 87 participants completed the study: 21 young healthy adults (YC), 21 older healthy adults (EC), 23 patients with Parkinson’s disease (PD), and 22 patients with Alzheimer’s disease (AD). Their demographic and clinical details are summarised in Table 2. Initially, 90 participants were recruited (22 YC, 21 EC, 25 PD, 22 AD); however, three individuals (1 YC and 2 PD) were excluded from all analyses due to technical issues during data acquisition.

All participants were recruited locally in Oxford, UK. None were professional artists, had a history of psychiatric illness, or were taking psychoactive medications (excluding standard dopamine replacement therapy for PD patients). Young participants were recruited via the University of Oxford Department of Experimental Psychology recruitment system. Older healthy volunteers (all >50 years of age) were recruited from the Oxford Dementia and Ageing Research (OxDARE) database.

Patients with PD were recruited from specialist clinics in Oxfordshire. All had a clinical diagnosis of idiopathic Parkinson's disease and no history of other major neurological or psychiatric conditions. While specific dosages of dopamine replacement therapy (e.g., levodopa equivalent doses) were not systematically recorded, all patients were tested while on their regular medication regimen ('ON' state).

Patients with PD were recruited from clinics in the Oxfordshire area. All had a clinical diagnosis of idiopathic Parkinson’s disease and no history of other major neurological or psychiatric illnesses. While all patients were tested in their regular medication ‘ON’ state, the specific pharmacological profiles—including the exact types of medication (e.g., levodopa, dopamine agonists, or combinations) and dosages—were not systematically recorded. The disease duration and PD severity were also un-recorded for this study.

Patients with AD were recruited from the Cognitive Disorders Clinic at the John Radcliffe Hospital, Oxford, UK. All AD participants presented with a progressive, multidomain, predominantly amnestic cognitive impairment. Clinical diagnoses were supported by structural MRI and FDG-PET imaging consistent with a clinical diagnosis of AD dementia (e.g., temporo-parietal atrophy and hypometabolism).69 All neuroimaging was reviewed independently by two senior neurologists (S.T. and M.H.).

Global cognitive function was assessed using the Addenbrooke’s Cognitive Examination-III (ACE-III).70 All healthy participants scored above the standard cut-off of 88, with the exception of one elderly participant who scored 85. In the PD group, two participants scored below the cut-off (85 and 79). In the AD group, six participants scored above 88; these individuals were included based on robust clinical and radiological evidence of AD pathology rather than their ACE-III score alone.”

(3) YA and OA patients appear to differ in gender distribution.

We acknowledge the difference in gender distribution between the young (71.4% female) and older adult (57.1% female) cohorts. However, we do not anticipate that gender influences the fundamental computational mechanisms of retinotopic maintenance or transsaccadic remapping. These processes represent low-level visuospatial functions for which there is no established evidence of gender-specific differences in precision or coordinate transformation. We have ensured that the gender distribution for each cohort is clearly listed in the demographics table (Table 2) for full transparency.

Thank you very much for very insightful feedback!

**Reviewer #3 (Public review):**

Thank you for the positive feedback regarding our inclusion of clinical groups and the identification of computational phenotypes that differentiate these cohorts.

To address your concerns about the model, we have clarified our use of Bayesian Model Selection, which inherently penalises model complexity to ensure that our results are not driven solely by the number of parameters. We will also provide further evidence regarding model generalisability to address the concern of overfitting.

Regarding the link with the ROCF, we have revised the manuscript to better highlight the specific relationship between our transsaccadic parameters and the ROCF data and better motivate the inclusion of these results in the main text.

Below is our response to your suggestions point-by-point:

(1) The models tested differ in terms of the number of parameters. In general, a larger number of parameters leads to a better goodness of fit. It is not clear how the difference in the number of parameters between the models was taken into account. It is not clear whether the modelling results could be influenced by overfitting (it is not clear how well the model can generalize to new observations).

To ensure our results were not driven by the number of parameters, we utilised random-effects Bayesian Model Selection (BMS) to adjudicate between our candidate models. Unlike maximum likelihood methods, BMS relies on the marginal likelihood (model evidence), which inherently balances model fit against parsimony—a principle known as the Occam’s Razor (Rasmussen and Ghahramani, 2000). In this framework, a model is only preferred if the improvement in fit justifies the additional parameter space; redundant parameters actually lower model evidence by diluting the probability mass. We would be happy to point toward literature that discusses how these marginal likelihood approximations provide a more robust guard against overfitting than standard metrics like BIC or AIC (MacKay, 2003; Murray and Ghahramani, 2005; Penny, 2012).

The fact that the "Dual (Saccade) + Interference" model (Model 7) emerged as the winner—with a Bayes Factor of 6.11 against the next best alternative—demonstrates that its complexity was statistically justified by its superior account of the trial-by-trial data.

Furthermore, to address the risk of overfitting, we established the generalisability of these parameters by using them to predict performance on an independent clinical task. These parameters successfully explained ~62% of the variance in ROCF copy scores—a very distinct, real-world task--confirming that they represent robust computational phenotypes rather than idiosyncratic fits to the initial dataset.

In the Results (p10):

“We used random-effects Bayesian model selection to identify the most plausible generative model. This process relies on the marginal likelihood (model evidence), which inherently balances model fit against complexity—a principle often referred to as Occam’s razor.[25–27]”

In the Discussion (p17):

“Importantly, the risk of overfitting is mitigated by the Bayesian Model Selection framework; by utilising the marginal likelihood for model comparison, the procedure inherently penalises excessive model complexity and promotes generalisability.[25–27,42] This generalisability was further evidenced by the model's ability to predict performance on the independent ROCF task, confirming that these parameters represent robust mechanistic phenotypes rather than idiosyncratic fits to the initial dataset.”

(2) Results specificity: it is not clear how specific the modelling results are with respect to constructional ability (measured via the Rey-Osterrieth Complex Figure test). As with any cognitive test, performance can also be influenced by general, non-specific abilities that contribute broadly to test success.

We agree that constructional performance is influenced by both specific mechanistic constraints and general cognitive abilities. To isolate the unique contribution of transsaccadic updating, we therefore performed a partial correlation analysis across the entire sample. We examined the relationship between location error in the two-saccades condition (our primary behavioural measure of transsaccadic memory) and ROCF copy scores. Even after partialling out the effects of global cognitive status (ACE-III total score), age, and years of education, the correlation remained highly significant (rho = -0.39, p < 0.001).

This suggests that our model captures a specific computational phenotype—the precision of spatial updating during active visual sampling—rather than acting as a proxy for non-specific cognitive decline. This mechanistic link explains why traditional working memory measures (e.g., digit span or Corsi blocks) frequently fail to predict drawing performance; unlike those tasks, figure copying requires thousands of saccades, making it uniquely sensitive to the precision of the dynamic remapping signals identified by our modelling framework.

We added the following text in the Discussion (p19):

“We also found that the relationship between transsaccadic working memory and ROCF performance remains highly significant (rho = -0.39, p < 0.001), even after controlling for age, education, and global cognitive status (ACE-III total score). Consequently, transsaccadic updating may represent a discrete computational phenotype required for visuomotor control, rather than a non-specific proxy for global cognitive decline.[57]”

**Reviewer #3 (Recommendations for the authors):**
(1) The authors mention in the introduction the following: "One key hypothesis is that we use working memory across visual fixations to update perception dynamically", citing the following manuscript:Harrison, W. J., Stead, I., Wallis, T. S. A., Bex, P. J. & Mattingley, J. B. A computational 906 account of transsaccadic attentional allocation based on visual gain fields. Proc. Natl. 907 Acad. Sci. U.S.A. 121, e2316608121 (2024).However, the manuscript above does not refer explicitly to the involvement of working memory in transaccadic integration of object location in space. Rather, it takes advantage of recent evidence showing how the true location of a visual object is represented in the activity of neurons in primary visual cortex (A. P. Morris, B. Krekelberg, A stable visual world in primate primary visual cortex. Curr. Biol. 29, 1471-1480.e6 (2019)). The model hypothesizes that true locations of objects are readily available, and then allocates attention in real-world coordinates, allowing efficient coordination of attention and saccadic eye movements.

Thank you for clarification. As suggested, we have now included the citation of Morris & Krekelberg (2019) to acknowledge the evidence for stable object locations within the primary visual cortex.

(2) The authors in the introduction and the title use the terms 'transaccadic memory' and 'spatial working memory'. However, it is not clear whether these can be used interchangeably or are reflecting different constructs.Classical measures of visuo-spatial working memory are derived from the Corsi task (or similar), where the location of multiple objects is displayed and subsequently remembered. In such tasks, eye movements and saccades are not generally considered, only memory performance, representing the visuo-spatial span.Transaccadic memory tasks are instead explicitly measuring the performance on remembered object locations of features across explicit eye movements, usually using a very limited number of objects (1 or 2, as is the case for the current manuscript).While the two constructs share some features, it is not clear whether they represent the same underlying ability or not, especially because in transaccadic tasks, participants are required to perform one or more saccades, thus representing a dual-task case.I think the relationship between 'transaccadic memory' and 'spatial working memory' should be clarified in the manuscript.

Thank you. Yes, we have added this within the Methods - Measurement of saccade cost to clarify that spatial working memory is the broad cognitive construct responsible for short-term maintenance, whereas transsaccadic memory is the specific, dynamic process of remapping representations to maintain stability across eye movements.

In Methods (p.22):

“Within this framework, it is important to distinguish between the broad construct of spatial working memory and the specific process of transsaccadic memory. While spatial working memory refers to the general ability to maintain spatial information over short intervals, transsaccadic memory describes the dynamic updating of these representations—termed remapping—to ensure stability across eye movements. Unlike classical 'static' measures of spatial working memory, such as the Corsi block task which focuses on memory span, transsaccadic memory tasks explicitly require the integration of stored visual information with motor signals from intervening saccades. Our paradigm treats transsaccadic updating as a core computational process within spatial working memory, where eye-centred representations are actively reconstructed based on noisy memories of the intervening saccade vectors.”

(3) In Figure 1, the second row indicates the presentation of item 2. Indeed, in the condition 'saccade-after-item-1', the target in the second row of Figure 1 is displaced, as expected. This clarifies the direction and amplitude of the first saccade requested. However, from Figure 1, it is hard to understand the amplitude and direction of the second requested saccade. I think the figure should be updated, giving a full description of the direction and amplitude of the second saccade as well ('saccade-after-item-2' and 'two-saccades' conditions).

We agree that making the figure legend more self-contained is beneficial for the reader. While the specific physical parameters and the trial sequence for each condition are detailed in the Results and Methods sections, we have now updated the legend for Figure 1 to explicitly define these details. Specifically, we have clarified that the colour wheel itself served as the target for the second instructed saccade (i.e., the movement from the second fixation cross to the colour wheel location). We have also included the quantitative constraint that all saccade vectors were at least 8.5 degrees of visual angle in amplitude. Given the limited space within a figure legend, we hope these concise additions provide the transparency requested without interrupting the conceptual flow of the diagram.

Updated Figure 1 legend:

“Participants were asked to fixate a white cross, wherever it appeared. They had to remember the colour and location of a sequence of two briefly presented coloured squares (Item 1 and 2), each appearing within a white square frame. They then fixated a colour wheel wherever it appeared on the screen, which served as the target for the second instructed saccade (i.e., a movement from the second fixation cross to the colour wheel location). This cued recall of a specific square (Item 1 or Item 2 labelled within the colour wheel). Participants selected the remembered colour on the colour wheel which led to a square of that colour appearing on the screen. They then dragged this square to its remembered location on the screen. Saccadic demands were manipulated by varying the locations of the second frame and the colour wheel, resulting in four conditions in their reliance on retinotopic versus transsaccadic memory: (1) No-Saccade condition providing a baseline measure of within-fixation precision as no eye movements were required. (2) Saccade After Item 1; (3) Saccade After Item 2; (4) Saccades after both items (Two Saccades condition). In all conditions requiring eye movements, saccade vectors were constrained to a minimum amplitude of 8.5° (degrees of visual angle). While the No-Saccade condition isolates retinotopic working memory, conditions (2) to (4) collectively quantify the impact of varying saccadic demands and timings on the maintenance of spatial information, thereby assessing the efficacy of the transsaccadic updating process.”

(4) The authors write: "Eye tracking analysis confirmed high compliance: participants correctly maintained fixation or executed saccades as instructed on the vast majority of trials (83% {plus minus} 14%). Non-compliant trials were excluded 136 from further analysis." 14% of excluded trials are a substantial fraction of trials, given the task requirements. Is this proportion of excluded trials different between experimental groups, and are experimental groups contributing equally to this proportion?

We thank the reviewer for pointing this out, and we apologise for the confusion. The 83% trial number was actually across all four cohorts, and all conditions, and it was actually above 90% for YC, EC and even AD, but dropped to 60 ish in PD group.

We now have conducted a full analysis of compliant trial counts using a mixed ANOVA (4 saccade conditions x 4 cohorts). This analysis revealed a main effect of group (F(3, 80) = 8.06, p < 0.001), which was driven by lower compliance in the PD cohort (mean approx. 25.4 trials per condition) compared to the AD, EC, and YC cohorts (means ranging from 35.8 to 38.9 trials per condition). Crucially, however, the interaction between group and condition was not statistically significant (p = 0.151). This indicates that the relative impact of saccade demands on trial retention was consistent across all four groups.

Because our primary behavioural measure—the saccade cost—is a within-subject comparison of impairment across conditions, these differences in absolute trial numbers do not introduce a systematic bias into our findings. Furthermore, even with the higher attrition in the PD group, we retained a sufficient number of high-quality trials (minimum mean of ~23 trials in the most demanding condition) to support robust trial-by-trial parameter estimation and valid statistical inference. We have updated the Results and Methods to reflect these details.

In Results (p4):

“To mitigate potential confounds, we monitored eye position throughout the experiment. Eye-tracking analysis confirmed high compliance in healthy adults, who followed instructions on the vast majority of trials (Younger Adults: 97.2 ± 5.2 %; Older Adults: 91.3 ± 20.4 %). The mean difference between these groups was negligible, representing just 1.25 trials per condition, and was not statistically significant (t(80) = 0.16, p = 1.000; see more in Methods – Eyetracking data analysis). Non-compliant trials were excluded from all further analyses.”

In Methods (p27):

“Eye-tracking analysis confirmed high compliance overall, with participants correctly maintaining fixation or executing saccades on the vast majority of trials (83% across all participants). A mixed ANOVA revealed a main effect of group on trial retention (F(3, 80) = 8.06, p < 0.001, partial η² = 0.23), primarily due to lower compliance in the PD cohort (YC: 97±4%; EC: 91±10%; AD: 95±5%; PD: 63±38%). Importantly, there was no significant interaction between group and saccade condition (F(3.36, 80) = 1.78, p = 0.15, partial η² = 0.008), suggesting that trial attrition was not disproportionately affected by specific task demands in any group.

We acknowledge that this reduced trial count in the PD group represents a limitation for across-cohort comparison. However, the absolute number of compliant trials in PD group (mean approx. 25 per condition) remained sufficient for robust trial-by-trial parameter estimation. Furthermore, the lack of a significant group-by-condition interaction confirms that the results reported for this cohort remain valid and that our primary finding of a selective spatial memory deficit is robust to these differences in data retention.”

(5) Modelling(a) Degrees of freedom, cross-validation, number of parameters.I appreciate the effort in introducing and testing different models. Models of increase in complexity and are based on different assumptions about the main drivers and mechanisms underlying the dependent variable. The models differ in the number of parameters. How are the differences in the number of parameters between models taken into account in the modelling analysis? Is there a cost associated with the extra parameters included in the more complex models?(b) Cross-validation and overfitting.Overfitting can occur when a model learns the training data but cannot generalize to novel datasets. Cross-validation is one approach that can be used to avoid overfitting. Was cross-validation (or other approaches) implemented in the fitting procedure against overfitting? Otherwise, the inference that can be derived from the modelled parameters can be limited.

To address your concerns regarding model complexity and overfitting, we would like to clarify our use of Bayesian Model Selection (BMS). Unlike frequentist methods that often rely on cross-validation to assess generalisability, we used random-effects BMS based on the marginal likelihood (model evidence). This approach inherently implements Bayesian Occam’s Razor by integrating out the parameters. Under this framework, the use of the marginal likelihood for model selection provides a mathematically equivalent safeguard to frequentist cross-validation, as it evaluates the model's ability to generalise across the entire parameter space rather than just finding a maximum likelihood fit for the training data. Thus, models are penalised not just for the absolute number of parameters, but for their overall functional flexibility. A more complex model is only preferred if the improvement in model fit is substantial enough to outweigh this inherent penalty. The emergence of Model 7 as the winner (Bayes Factor = 6.11 against the next best alternative) confirms that its additional complexity is statistically justified.

Furthermore, in this study we provided an external validation of these recovered parameters by demonstrating that they explain 62% of the variance in an independent, real-world, clinical task (ROCF copy). This empirical evidence confirms that our model captures robust mechanistic phenotypes rather than idiosyncratic noise. We have updated the Results and Discussion to explicitly state these.

In Results: (p10)

“We used random-effects Bayesian model selection to identify the most plausible generative model. This process relies on the marginal likelihood (model evidence), which inherently balances model fit against complexity—a principle often referred to as Occam’s razor.[26–28]”

In Discussion: (p17)

“Importantly, the risk of overfitting is mitigated by the Bayesian Model Selection framework; by utilising the marginal likelihood for model comparison, the procedure inherently penalises excessive model complexity and promotes generalisability.[26–28,43] This generalisability was further evidenced by the model's ability to predict performance on the independent ROCF task, confirming that these parameters represent robust mechanistic phenotypes rather than idiosyncratic fits to the initial dataset.”

(6) n. of participants.(a) The authors write the following: "A total of healthy volunteers (21 young adults, mean age = 24.1 years; 21 older adults, mean age = 72.4 years) participated in this study. Their demographics are shown in Table 1. All participants were recruited locally in Oxford." However, Table 1 reports the data from more than 80 participants, divided into 4 groups. Details about the PD and AD groups are missing. Please clarify.

We apologize for this lack of clarity in the text. We have rewrote and expand the “Participants” section and corrected Table 2 in the Methods section to reflect the correct number of participants.

In Methods (p20):

“A total of 87 participants completed the study: 21 young healthy adults (YC), 21 older healthy adults (EC), 23 patients with Parkinson’s disease (PD), and 22 patients with Alzheimer’s disease (AD). Their demographic and clinical details are summarised in Table 2. Initially, 90 participants were recruited (22 YC, 21 EC, 25 PD, 22 AD); however, three individuals (1 YC and 2 PD) were excluded from all analyses due to technical issues during data acquisition.

All participants were recruited locally in Oxford, UK. None were professional artists, had a history of psychiatric illness, or were taking psychoactive medications (excluding standard dopamine replacement therapy for PD patients). Young participants were recruited via the University of Oxford Department of Experimental Psychology recruitment system. Older healthy volunteers (all >50 years of age) were recruited from the Oxford Dementia and Ageing Research (OxDARE) database.

Patients with PD were recruited from specialist clinics in Oxfordshire. All had a clinical diagnosis of idiopathic Parkinson's disease and no history of other major neurological or psychiatric conditions. While specific dosages of dopamine replacement therapy (e.g., levodopa equivalent doses) were not systematically recorded, all patients were tested while on their regular medication regimen ('ON' state).

Patients with PD were recruited from clinics in the Oxfordshire area. All had a clinical diagnosis of idiopathic Parkinson’s disease and no history of other major neurological or psychiatric illnesses. While all patients were tested in their regular medication ‘ON’ state, the specific pharmacological profiles—including the exact types of medication (e.g., levodopa, dopamine agonists, or combinations) and dosages—were not systematically recorded. The disease duration and PD severity were also un-recorded for this study.

Patients with AD were recruited from the Cognitive Disorders Clinic at the John Radcliffe Hospital, Oxford, UK. All AD participants presented with a progressive, multidomain, predominantly amnestic cognitive impairment. Clinical diagnoses were supported by structural MRI and FDG-PET imaging consistent with a clinical diagnosis of AD dementia (e.g., temporo-parietal atrophy and hypometabolism).[70] All neuroimaging was reviewed independently by two senior neurologists (S.T. and M.H.).

Global cognitive function was assessed using the Addenbrooke’s Cognitive Examination-III (ACE-III).[71] All healthy participants scored above the standard cut-off of 88, with the exception of one elderly participant who scored 85. In the PD group, two participants scored below the cut-off (85 and 79). In the AD group, six participants scored above 88; these individuals were included based on robust clinical and radiological evidence of AD pathology rather than their ACE-III score alone.”

(b) As modelling results rely heavily on the quality of eye movements and eye traces, I believe it is necessary to report details about eye movement calibration quality and eye traces quality for the 4 experimental groups, as noisier data could be expected from naïve and possibly older participants, especially in case of clinical conditions. Potential differences in quality between groups should be discussed in light of the results obtained and whether these could contribute to the observed patterns.

Thank you for pointing this out. We have revised the Methods about how calibration was done:

(p27) “Prior to the experiment, a standard nine-point calibration and validation procedure was performed. Participants were instructed to fixate a small black circle with a white centre (0.5 degrees) as it appeared sequentially at nine points forming a 3 x 3 grid across the screen. Calibration was accepted only if the mean validation error was below 0.5 degrees and the maximum error at any single point was below 1.0 degree. If these criteria were not met, or if the experimenter noticed significant gaze drift between blocks, the calibration procedure was repeated. This calibration ensured high spatial accuracy across the entire display area, facilitating the precise monitoring of fixations on item frames and saccadic movements to the response colour wheel.”

Moreover, as detailed in our response to Point 4, while the PD group exhibited lower compliance, there was no interaction between group and saccade condition for compliance (p = 0.151). This confirms that any noise or trial attrition was distributed evenly across experimental conditions. Consequently, the observed "saccade cost" (the difference in error between conditions) is not an artefact of unequal noise but represents a genuine mechanistic impairment in spatial updating. We have updated the Methods to clarify this distinction.

Furthermore, our Bayesian framework explicitly estimates precision (random noise) as a distinct parameter from updating cost (saccade cost). This allows the model to partition the variance: even if a clinical group is "noisier" overall, this is captured by the precision parameter, ensuring it does not inflate the specific estimate of saccade-driven memory impairment.

(7) Figure 5. I suggest reporting these results using boxplots instead of barplots, as the former gives a better overview of the distributions.

We appreciate the suggestion to use boxplots to better illustrate data distributions. However, we have chosen to retain the current bar plot format due to the visual and statistical complexity of our 4 x 4 x 2 experimental design. Figure 5 represents 16 distinct distributions across four groups and four conditions for both location and colour measures; employing boxplots/violins for this density of data would significantly increase visual clutter and make the figure difficult to parse.

Furthermore, the primary objective of this figure is to reflect the statistical analysis and illustrate group differences in overall performance and highlight the specific finding that patients with AD were significantly more impaired across all conditions compared to YC, EC, and PD groups. Our statistical focus remains on the mean effects—specifically the significant main effect of group (F(3, 318) = 59.71, p < 0.001) and the critical null-interaction between group and condition (p = 0.90). The error measure most relevant to these comparisons is the standard error of the mean (SEM), rather than the interquartile range (IQR). We think that bar plots provide the most straightforward and scannable representation of these mean differences and the consistent pattern of decay across cohorts for the final manuscript layout.

To address the reviewer’s request for distributional transparency, we have provided a version of Figure 5 using grouped boxplots in the supplementary material (Supplementary figure 2). We note, however, that the spread of raw data points in these plots does not directly reflect the variance associated with our within-subject statistical comparisons.

(8) Results specificity, trans-saccadic integration and ROCF. The authors demonstrate that the derived model parameters account for a significant amount of variability in ROCF performance across the experimental groups tested (Figure 8A). However, it remains unclear how specific the modelling results are with respect to the ROCF.The ROCF is generally interpreted as a measure of constructional ability. Nevertheless, as with any cognitive test, performance can also be influenced by more general, non-specific abilities that contribute broadly to test success. To more clearly link the specificity between modelling results and constructional ability, it would be helpful to include a test measure for which the model parameters would not be expected to explain performance, for example, a verbal working memory task.I am not necessarily suggesting that new data should be collected. However, I believe that the issue of specificity should be acknowledged and discussed as a potential limitation in the current context.

We appreciate this important point regarding the discriminant validity of our findings. We agree that cognitive performance in clinical populations is often influenced by a general "g-factor" or non-specific executive decline. However, we chose the ROCF Copy task specifically because it is a hallmark clinical measure of constructional ability that effectively serves as a real-world transsaccadic task, requiring participants to integrate spatial information across hundreds of saccades between the model figure and the drawing surface.

To address the reviewer’s concern regarding specificity, we leveraged the fact that all participants completed the ACE-III, which includes a dedicated verbal memory component (the ACE Memory subscale). We conducted a partial correlation analysis and found that the relationship between transsaccadic working memory and ROCF copy performance remains highly significant (rho = -0.46, p < 0.001), even after controlling for age, education, and the ACE-III Memory subscale score. This suggests that the link between transsaccadic updating and constructional ability is mechanistically specific rather than a byproduct of global cognitive impairment. We have substantially revised the Discussion to highlight this link and the supporting statistical evidence.

We first updated the last paragraph of Introduction:

“Finally, by linking these mechanistic parameters to a standard clinical measure of constructional ability (the Rey-Osterrieth Complex Figure task), we demonstrate that transsaccadic updating represents a core computational phenotype underpinning real-world visuospatial construction in both health and neurodegeneration.”

The new section in Discussion highlighting the ROCF copy link:

“Importantly, our computational framework establishes a direct mechanistic link between trassaccadic updating and real-world constructional ability. Specifically, higher saccade and angular encoding errors contribute to poorer ROCF copy scores. By mapping these mechanistic estimates onto clinical scores, we found that the parameters derived from our winning model explain approximately 62% of the variance in constructional performance across groups. These findings suggest that the computational parameters identified in the LOCUS task represent core phenotypes of visuospatial ability, providing a mechanistic bridge between basic cognitive theory and clinical presentation.

This relationship provides novel insights into the cognitive processes underlying drawing, specifically highlighting the role of transsaccadic working memory. Previous research has primarily focused on the roles of fine motor control and eye-hand coordination in this skill.[4,50–55] This is partly because of consistent failure to find a strong relation between traditional memory measures and copying ability.[4,31] For instance, common measures of working memory, such as digit span and Corsi block tasks, do not directly predict ROCF copying performance.[31,56] Furthermore, in patients with constructional apraxia, these memory performance often remain relatively preserved despite significant drawing impairments.[56–58] In literature, this lack of association has often been attributed to “deictic” visual-sampling strategies, characterised by frequent eye movements that treat the environment as an external memory buffer, thereby minimising the need to maintain a detailed internal representation.[4,59] In a real-world copying task, the ROCF requires a high volume of saccades, making it uniquely sensitive to the precision of the dynamic remapping signals identified here. Recent eye-tracking evidence confirms that patients with AD exhibit significantly more saccades and longer fixations during figure copying compared to controls, potentially as a compensatory response to trassaccadic working memory constraints.[56] This high-frequency sampling—averaging between 150 and 260 saccades for AD patients compared to approximately 100 for healthy controls—renders the task highly dependent on the precision of dynamic remapping signals.[56] We also found that the relationship between transsaccadic working memory and ROCF performance remains highly significant (rho = -0.46, p < 0.001), even after controlling for age, education, and ACE-III Memory subscore. Consequently, transsaccadic updating may represent a discrete computational phenotype required for visuomotor control, rather than a non-specific proxy for global cognitive decline.[58]

In other words, even when visual information is readily available in the world, the act of drawing performance depends critically on working memory across saccades. This reveals a fundamental computational trade-off: while active sampling strategies (characterised with frequent eye-hand movements) effectively reduce the load on capacity-limited working memory, they simultaneously increase the demand for precise spatial updating across eye movements. By treating the external world as an "outside" memory buffer, the brain minimises the volume of information it must hold internally, but it becomes entirely dependent on the reliability with which that information is remapped after each eye movement. This perspective aligns with, rather contradicts, the traditional view of active sampling, which posits that individuals adapt their gaze and memory strategies based on specific task demands.[3,60] Furthermore, this perspective provides a mechanistic framework for understanding constructional apraxia; in these clinical populations, the impairment may not lie in a reduced memory "span," but rather in the cumulative noise introduced by the constant spatial remapping required during the copying process.[58,61]

Beyond constructional ability, these findings suggest that the primary evolutionary utility of high-resolution spatial remapping lies in the service of action rather than perception. While spatial remapping is often invoked to explain perceptual stability,[11–13,15] the necessity of high-resolution transsaccadic memory for basic visual perception is debated.[13,62–64] A prevailing view suggests that detailed internal models are unnecessary for perception, given the continuous availability of visual information in the external world.[13,44] Our findings support an alternative perspective, aligning with the proposal that high-resolution transsaccadic memory primarily serves action rather than perception.[13] This is consistent with the need for precise localisation in eye-hand coordination tasks such as pointing or grasping.[65] Even when unaware of intrasaccadic target displacements, individuals rapidly adjust their reaching movements, suggesting direct access of the motor system to remapping signals.[66] Further support comes from evidence that pointing to remembered locations is biased by changes in eye position,[67] and that remapping neurons reside within the dorsal “action” visual pathway, rather than the ventral “perception” visual pathway.[13,68,69] By demonstrating a strong link between transsaccadic working memory and drawing (a complex fine motor skill), our findings suggest that precise visual working memory across eye movements plays an important role in complex fine motor control.”

We are deeply grateful to the reviewers for their meticulous reading of our manuscript and for the constructive feedback provided throughout this process. Your insights have significantly enhanced the clarity and rigour of our work.

In addition to the changes requested by the reviewers, we wish to acknowledge a reporting error identified during the revision process. In the original Results section, the repeated measures ANOVA statistics for YC included Greenhouse-Geisser corrections, and the between-subjects degrees of freedom were incorrectly reported as within-subjects residuals. Upon re-evaluation of the data, we confirmed that the assumption of sphericity was not violated; therefore, we have removed the unnecessary Greenhouse-Geisser corrections and corrected the degrees of freedom throughout the Results and Methods sections. We have ensured that these statistical updates are reflected accurately in the revised manuscript and that they do not alter the significance or interpretation of any of our primary findings.

We hope that these revisions address all the concerns raised and provide a more robust account of our findings. We look forward to your further assessment of our work.